# Targeting SMAD-Dependent Signaling: Considerations in Epithelial and Mesenchymal Solid Tumors

**DOI:** 10.3390/ph17030326

**Published:** 2024-03-01

**Authors:** Farhana Runa, Gabriela Ortiz-Soto, Natan Roberto de Barros, Jonathan A. Kelber

**Affiliations:** 1Department of Biology, California State University Northridge, Northridge, CA 91330, USA; fruna2014@gmail.com; 2Department of Biology, Baylor University, Waco, TX 76706, USA; gabriela_ortizsoto@baylor.edu; 3Terasaki Institute for Biomedical Innovation, Los Angeles, CA 90024, USA; nbarros@terasaki.org

**Keywords:** Transforming Growth Factor-β (TGF-β), SMADs, Epithelial–Mesenchymal Transition (EMT), Inhibitory SMADs (I-SMADs), non-canonical SMAD signals, nucleocytoplasmic trafficking, small-molecule inhibitors

## Abstract

SMADs are the canonical intracellular effector proteins of the TGF-β (transforming growth factor-β). SMADs translocate from plasma membrane receptors to the nucleus regulated by many SMAD-interacting proteins through phosphorylation and other post-translational modifications that govern their nucleocytoplasmic shuttling and subsequent transcriptional activity. The signaling pathway of TGF-β/SMAD exhibits both tumor-suppressing and tumor-promoting phenotypes in epithelial-derived solid tumors. Collectively, the pleiotropic nature of TGF-β/SMAD signaling presents significant challenges for the development of effective cancer therapies. Here, we review preclinical studies that evaluate the efficacy of inhibitors targeting major SMAD-regulating and/or -interacting proteins, particularly enzymes that may play important roles in epithelial or mesenchymal compartments within solid tumors.

## 1. Introduction

Transforming growth factor-β (TGF-β) superfamily and resulting canonical SMAD signaling have gained significant attention in cancer research since the mid-1980s because of their significance in regulating central functions of the cell, such as proliferation, apoptosis, adhesion, and differentiation. SMADs specifically function to transmit information from extracellular signals received by TGF-β receptors, registering at the plasma membrane to the nucleus downstream of TGF-β. In the nucleus, SMADs cooperate with transcription factors, co-activators, and co-repressor regulators to control gene expression in a context-dependent manner [1,2]. Importantly, many non-SMAD factors are involved in directly controlling how SMAD proteins function in the TGF-β pathway to support, attenuate, or modulate downstream cellular responses.

TGF-β/SMAD plays a biphasic function during tumor progression, where it can suppress or potentiate tumorigenesis in normal and pre-malignant epithelial cells [3]. Signals downstream of TGF-β produced by tumors in the TME (tumor microenvironment) can also activate tumors to undergo an EMT (epithelial–mesenchymal transition) and/or hybrid/partial EMT [4,5,6,7]. EMT is well known to confer invasive, therapy-resistant, and metastatic properties to cancer cells [8,9,10]. Since TGF-β signaling plays a critical role in tumor progression, targeting the downstream SMAD signals presents an enormous challenge in the effort to develop target therapies to eradicate solid tumors effectively. Previous reviews have highlighted novel approaches to targeting signals downstream of TGF-β that focus on microRNAs (miRNAs), long non-coding RNAs (lnRNAs), deubiquitinating enzymes (DUBs), and protein–protein interactions (PPIs) [11,12,13,14,15]. However, the pleiotropic nature of TGF-β signaling contributes to alternate pathway engagement and tumor escape and drug resistance, creating challenges for clinicians and patients. Preclinical findings demonstrate that the TGF-β pathway can be modulated and/or targeted potentially using different alternatives. Although mouse/human chimeric monoclonal antibodies, receptors ligand traps, synthetic DNA antisense oligomers, therapeutic vaccines, and small-molecule inhibitors [16,17,18,19] are worth mentioning, these therapies commonly disrupt normal physiological functions and provide unsatisfactory results in clinical trials [20,21].

Here, we present a consideration of preclinical studies testing the anti-tumor efficacy of pharmacological targeting SMAD-regulating and/or -interacting proteins. We summarize inhibitors targeting the major SMAD-interacting enzymes involved in each pathway in addition to non-SMAD non-canonical pathways downstream of the TGF-β signaling. Finally, we propose how these inhibitors may contribute to abrogating cancer progression induced by TGF-β/SMAD signaling in the epithelial and/or mesenchymal cell types that are commonly found within the solid TME.

## 2. Canonical TGF-β/SMAD Signaling

The basic principles underlying TGF-β signaling are well-founded. Ligands within the TGF-β superfamily are known as activins/NODALs, bone morphogenetic proteins (BMPs), and growth differentiation factors (GDFs). Ligand-driven signaling activity depends on the formation of pairs with type I and type II receptors. The receptors (serine/threonine kinase) take part in the formation of dimers with disulfide-linked ligand dimers [22,23]. Upon their activation, the type I receptors are trans-phosphorylated by the type II receptors in the juxta membrane region (TTSGSGSG), which drives activation of the type I receptor kinase domain [24,25,26]. These activated receptor complexes subsequently activate a series of downstream signal transduction proteins known as SMAD proteins via phosphorylation that are transported to the nucleus, regulating various transcriptional programs.

Among the eight SMADs in vertebrates, only five are recognized as R-SMADs (Receptor-SMADs). TGF-β- and activins/NODAL-mediated signals activate SMAD2 and SMAD3 predominantly, whereas GDFs and BMPs initiate the activation of SMAD1, SMAD5, and SMAD8 [27]. Two serines are involved to initiate the phosphorylation of receptor and R-SMADs. Primarily, the phosphorylation event of R-SMADs initiates them to form homomeric complexes and subsequently primes them to form complexes with R-SMADs and SMAD4 (Co-SMAD). After activation, the heteromeric complexes (R-SMADs-SMAD4) finally accumulate in the nucleus and bind to DNA (high-affinity sequences) cooperatively with transcription factors and/or co-factors and co-repressors to regulate transcription (Figure 1) [1,2].

## 3. Activity and Nucleocytoplasmic Trafficking of SMADS

### Mechanisms of SMAD Protein Nucleocytoplasmic Trafficking

SMAD proteins exist in dynamic states in the cytoplasm or nucleus with or without the stimulation of TGF-β, respectively. A study by Nicolás et al. [29] showed that both uninduced and induced cells with TGF-β could exhibit variation in SMAD distribution. Accumulating evidence supports that a few proteins are involved in the activation, retention, and trafficking of SMADs from the cytoplasm to the nucleus and vice versa (Figure 2). Activation begins with phosphorylation of the GS region on the type I receptor TGF-β (TGF-βRI) by the type II receptor (TGF-βRII), which leads to the affinity of receptor I for sxsmotif of R-SMADs at the C-terminal [30]. Thereby, the two C-terminal serines of R-SMADs are phosphorylated with substantial interaction between the R-SMADs and the type I receptor. This event is critical to changing the conformation of R-SMADs and allowing them to dissociate consequently from the receptor complex [31]. In this context, the association of the endocytic protein SARA (SMAD Anchor of Receptor Activation) [32,33] and a few SARA-associated proteins (e.g., cPML [34,35], endofin [32], and Dab2 [36] promote R-SMAD activation by recruiting them to the TGF-β receptor. Phosphorylated R-SMADs can interact with SMAD4 and form heteromeric complexes after dissociation from the TGF-β receptor complexes and subsequently transport to the nucleus [35,36,37,38]. It was also found that hepatocyte growth factor-regulated tyrosine kinase substrate (Hgs) is associated with SARA and promotes phosphorylation of SMADs 1, 5, and 8 induced by BMP signals [39,40]. The conformational change driven by the phosphorylation of R-SMAD monomers supports the dissociation from the receptor complex as well as promotes the interaction between the R-SMAD, which is phosphorylated at the C-terminal in addition to SMAD4 and another R-SMAD. Thus, different compositions of SMAD trimeric complexes can exist as a complex of R-SMAD monomers (activated homo or heterotrimer) and/or a complex of phosphorylated one or two R-SMADs with SMAD4 in the cytoplasm [41,42,43,44].

Nuclear translocation of SMAD complexes is sustained by some nuclear pore proteins. Although MH1 domains of SMAD proteins contain NLS-motifs, the regulation of trafficking of SMADs is governed by divergent methods including importin/exportin-mediated, Ran-dependent (Figure 2(Da)); phospho-dependent (Figure 2(Db,Dc)); importin-mediated Ran-independent (Figure 2(Dc)); and importin-exportin-independent and phospho-independent method (Figure 2(Dd)). Many importin/exportin family members and Ran proteins (small Ran GTPase) are involved in the trafficking of R-SMADs. Importin-β1 interacts at the MH1 domain of SMAD3 and is frequently imported into the nucleus through Ran-depended proteins [45], whereas exportin 4, which has a conserved sequence on the MH2 domain of SMAD3, allows the exportation of SMAD3 from the nucleus in a GTPase manner [46]. Importantly, RanBP3 acts as a selective nuclear protein of SMAD2/SMAD3 in the TGF-β pathway. In mammalian cells (human keratinocyte cells, HaCaT, human HEK293T, human HepG2, mouse myoblast cells C2C12) and *Xenopus* embryos, RanBP3 can directly identify dephosphorylated SMAD2/SMAD3 produced by the phosphatase activity of nuclear SMAD phosphatase. Thereby, the nuclear export of SMAD2/SMAD3 is augmented in a Ran-dependent manner to terminate the TGF-β signal [47]. On the other hand, importin-α mediates the nuclear translocation of SMAD4 [48]. Moreover, nuclear pore proteins (CAN/Nup214 and Nup15) can regulate the cytoplasmic shuttling of SMAD2, SMAD3, and SMAD4 proteins [49,50]. Further, dynein light chain km23-1 and Km23-2 can interact with SMAD2 and SMAD3, which facilitates the nuclear trafficking of the R-SMADS from cytoplasm to the nucleus for target gene transcription [51,52].

Other proteins, like AKT/PKB [53,54], can directly interact with SMAD3 and inhibit its phosphorylation and nuclear accumulation, further inhibiting SMAD3-mediated transcription and apoptosis. On the other hand, ERK/MAP [55,56] CDKs (2/4) [57,58] and GSKB3b [59,60] can inhibit the nuclear accumulation of R-SMADS by phosphorylating the linker region of R-SMADs in both epithelial and fibroblast cells. Nucleocytoplasmic shuttling is mediated by dephosphorylation, which promotes R-SMAD traveling from the nucleus to the cytoplasm. It was found that PPM1A functions as a phosphatase and terminates TGF-β signaling [61]. A study by Dai et al. [62] shows that PPM1A targets the nuclear exporter RanBp3 and thereby controls the transport of SMAD2/SMAD3 from the nucleus to the cytoplasm. RanBP3 is dephosphorylated at Ser 58, which promotes SMAD2/SMAD3 exportation from the nucleus. Thus, PPM1A provides the maximum RanBP3 exporter function for the effective termination of canonical TGF-β signaling.

Intriguingly, it was also found in epithelial, fibroblast, cancer (Hela cells, small lung cancer cell lines, breast cancer cell lines), and embryonic stem cells where SMADs can be reserved in the nucleus through the interaction with high-affinity DNA-binding transcription factors. YAP (Yes-associated protein), co-factors like Fast1/FoxH1, and TAZ (transcriptional co-activator with PDZ-binding motif) can associate with SMADs [63,64,65,66,67]. The nuclear retention inhibits the export of SMADs from the nucleus to the cytoplasm, and it is speculated that the association of YAP and SMAD3 in the nucleus may prevent the interaction of SMAD3 with exportin 4 or RanBP3. Further, it was found that phospho-SMAD-3 in epithelial cells can block the interaction of SMAD-4 with exportin 1 and thereby promote nuclear accumulation of SMAD4 [68].

Inhibitors targeting the SMAD-interacting enzymes such as AKT/PKB (Ser/Thr kinase/protein kinase B), CDK (cyclin-dependent kinase), GSK GSK3b (axin/glycogen synthase kinase 3 beta), and ERK (extracellular signal-related kinase) are cataloged in Table 1 [69,70,71,72,73,74,75,76,77,78,79,80,81,82,83]. These inhibitors act by modulating the phosphorylation activity of the kinases and thereby regulate their interaction, the R-SMADs, and SMAD-mediated signaling in epithelial and mesenchymal cells. Significant advancements have been observed in the area of small-molecule inhibitors of AKT functions through different mechanisms. In addition, pleckstrin homology domain, ATP-competitive, and allosteric inhibitors of AKT are noteworthy in developing clinical trials, and therapeutic benefits in treating different solid tumors have been observed [83]. A preclinical study of MK-2206 in epithelial morphology of pancreatic cancer cells by Wang et al. [72] showed that AKT phosphorylation is inhibited, and cell proliferation is attenuated by the application of the MK-2206 inhibitor. Further, a combination of MK-2206 with gemcitabine enhanced the inhibition of cancer cell proliferation. It suggests that gemcitabine-mediated AKT activation induced increased cell proliferation [72]. In epithelial cells, AKT can act as one of the major mediators of cell survival and apoptosis and regulates the activity of SMAD3 (Figure 2). In epithelioid pancreatic cancer cells, AKT phosphorylation might enhance the activity of SMAD3 and stimulate SMAD-mediated cell proliferation. It is considered that the inhibition of AKT-phosphorylation by MK-2206 may promote the interaction of AKT with SMAD3 and thereby support the inhibition of SMAD3 activation, nuclear accumulation, SMAD3-mediated transcription, and cell proliferation.

Another potential inhibitor is nitazoxanide (NTZ), which targets CDK1 by inhibiting phosphorylation of CDK1 at Thr161, as studied by Huang et al. [74]. It is thought that CDK1 phosphorylation might enhance the activity of SMAD3 and nucleocytoplasmic trafficking and stimulate SMAD-mediated cell proliferation in epithelioid glioblastoma cells. However, NTZ application might modulate SMAD-3 and CDK1 interaction, resulting in inhibition of SMAD3 activation, nucleocytoplasmic trafficking, transcription, and SMAD3-induced cell proliferation. Thus, modulation of the SMAD signaling by inhibiting the phosphorylation of kinase enzymes suggests that NTZ can be a promising CDK1 inhibitor for the development of clinical trials.

## 4. Regulation of SMAD Activity by Post-Translational Modifications (PTMs)

### 4.1. Phosphorylation and Dephosphorylation

Activation of R-SMADs is originated by phosphorylation induced by type I receptors of two serines in their carboxy SSXS motif induced by ligands. R-SMAD2 and R-SMAD3 are phosphorylated by the TβRI/ALK-5 (TGF-β-specific type I receptor); R-SMAD 1, 5, and 8 are phosphorylated by ALK-3/BMPRIA (BMP-specific type I receptor, BMP receptor IA). In addition, the structure of R-SMADs with the MHI and MH2 domains and linker region contains substrate sites for other kinases such as Map kinases, cyclin-dependent kinases (CDKs), extracellular signal-regulated kinase 1 (ERK1), and monopolar spindle kinase (MPS1), which regulate the stability of R-SMADs and subsequent transcriptional responses [84,85]. For example, ERK (extracellular signal-regulated kinase 1) phosphorylates SMAD2 on their MH1 domain and enhances SMAD-mediated transcription in epithelial cells [86]. Not all kinases that target R-SMADs potentiate SMAD-mediated transcription. In epithelium and mesenchymal cells, for example, PKC (protein kinase C) directly phosphorylates SMAD3, but this action inhibits SMAD3-dependent transcription [87]. Still, other R-SMAD phosphorylation events generate a platform for R-SMAD degradation via ubiquitin-dependent processes. A study by Saura et al. [88] shows that PKG1 (cGMP-mediated protein kinase 1)-mediated phosphorylation of SMAD3 at the MH1 domain fosters proteasomal degradation of SMAD3 in endothelial cells. Other small GTPase proteins like Ras promote ubiquitin-dependent SMAD2/3 degradation and stabilization of TβRII via the degradation of the SPSB1 protein [89]. Ras interacts and colocalizes with the SPSB (TβRII negative regulator) on the cell membrane, resulting in SPSB1 degradation [89]. Both ubiquitylation and degradation can also occur through phosphorylation of SMAD3 and SMAD2 by casein kinase 1 gamma 2 (CKIg2) and PAK4 [90,91]. And phosphorylation of SMAD2 by PAK2 confers the inactivation of SMAD2 by interfering with TGFβRI-SMAD2 interaction [92] in epithelial cells. SMAD4 is constitutively phosphorylated in epithelial cells (mink lung epithelial cells, Mv1Lu) and cancer cells (human squamous carcinoma cell lines, HSC-4), as shown by a study by Nakao et al. [93]. Although constitutive phosphorylation of SMAD4 has been observed in epithelial and cancer cells, the site of phosphorylation of SMAD4 remains to be identified. ERK promotes SMAD4 nuclear accumulation and enhances SMAD4-mediated transcriptional activity [94]. Other kinases, such as LKB1 (liver kinase B1) and MAP38, phosphorylate SMAD4 on MH1 and MH2 domains, respectively. LBK1 promotes SMAD4 stability by associating with a scaffolding protein LIPI and enhances the SMAD-dependent transcription as a negative regulator in controlling TGFβ gene responses and EMT [95,96]. On the other hand, MAP38-mediated phosphorylation at Ser^343^ of SMAD4 drives positive regulation of transcription in response to TGF-β-mediated apoptosis and cell proliferation in a kinase-dependent manner [96].

The dephosphorylation of C-terminal serine plays a significant role in deactivating the R-SMADs and mitigating the SMAD-mediated signaling. PPM1A (PP2Cα), designated as Mg^+^/Mn^+^-dependent 1A protein phosphatase, was the first phosphatase identified by Lin et al. [61] that dephosphorylates the SMAD2/3 at the C-terminus SXS motif. The phosphate MTMR4 dephosphorylates the activated SMAD2 and SMAD3 in the endosome [97]. Phosphates such as SCP1, SCP2, and SCP3 can remove linker phosphorylation at specific levels without interfering with phosphorylation at the C-terminal of SMAD2/SMAD3 to enhance the TGF-β signal [98]. On the other hand, in mammal keratinocyte cells and *Xenopus* embryos, dephosphorylation of SMAD1 by SCP1, SCP2, and SCP3 can attenuate the BMP signal [99].

### 4.2. Ubiquitylation and Deubiquitylation

Ubiquitylation involves the sequential covalent modification of protein substrates with ubiquitin molecules via the action of E1, E2, and E3 ubiquitin ligases, which mark the substrate for further activity or degradation. Regulation of R-SMADs is coordinated by ubiquitylation and ubiquitin-like modifications, which supports SMADs’ stability and subsequent activity and subcellular localization. Several ubiquitin ligases have been involved in the degradation of R-SMADs. HECT (the homologous E6-AP carboxy terminal) family E3 ligases SMURF1 and SMURF2 [100,101,102], NEED4-2/NEDD4L [103], Hsc70-interacting protein (CHIP) [104], and the RING-finger family E3 ligase SCF complexes such as Rbx1, Skp1, Cullins, F-box proteins [105,106] and Arkadia (also known as RNF111) [107,108] are mentioned here. In brief, SMURF1 promotes ubiquitylation of the R-SMADs SMAD1 and SMAD5 and indicates them for degradation [102]. SMURF2 polyubiquitylates SMAD2 and promotes degradation, whereas SMURF2 monoubiquitylates SMAD3, inhibiting the formation of SMAD3 complex [101,102]. NEDD4-2/NEDD4L ubiquitylates both SMAD2 and SMAD3, while CHIP, SCF (Skp1), and GSK3β only trigger SMAD3 ubiquitylation and degradation after phosphorylation [107]. In addition, Akadia triggers both phospho-SMAD2 and SMAD3 ubiquitylation and proteasomal degradation [108].

Monoubiquitylation also regulates the SMAD4-mediated signaling, and it is worth mentioning that SCF^skp2^ E3 ligase ubiquitylates SMAD4 at the MH2 domain [109], enabling SMAD4 mutations in cancer to be degraded. The tripartite motif-containing 33 E3 ligase (TRIM33/TIF1γ/Ectodermin) promotes the disruption of the SMAD complex by SMAD4 monoubiquitylation [110,111]. A deubiquitylating enzyme (DUB) can oppose the action of ubiquitylation on SMADs. For example, USP15 (ubiquitin-specific peptidase) removes the monoubiquitylation of SMAD3 and thereby enhances the access of SMAD complexes to its target promoter domains/complexes [112]. Further, DUB, ubiquitin-specific protease 9x (USP9X/FAM), can reverse the function of monoubiquitylation of SMAD4 at Lys^519^ and inhibit the association of SMAD4 with R-SMADs [113].

### 4.3. Acetylation, ADP-Ribosylation, and Sumoylation

In addition to phosphorylation and ubiquitylation, acetylation of R-SMADs marks the TGF-β-induced interaction of the transcription co-activators cAMP-response element-binding (CREB) protein (CBP) and p300 by acetyltransferase. SMAD2 is acetylated at the MH1 domain, whereas the MH2 domain of SMAD3 is primarily acetylated, and each acetylation event enhances SMAD-mediated transcription [114,115,116]. ADP-ribosylation is another type of PTM; thereby, one or more ADP-ribose moieties are added to arginine by ADP-ribosyl transferase to control cell signaling and various cellular processes. PRAP1 (ADP-ribose polymerase 1) induces ribosylation of SMAD3 in its MH1 domain and thereby dissociates SMAD3 from the SMAD3/SMAD4 complex. SMAD-mediated TGF-β responses are attenuated by the action PARP-1. Further, overexpression of PARP-1 promotes impaired SMAD3-mediated gene expression and EMT [117]. By SUMOylating, a small ubiquitin-like modifier (SUMO) polypeptide is covalently attached to the target protein. SUMO does not lead to the degradation of the proteins. It was found that SMAD3 was associated with a protein inhibitor of activated STATy (PIAS4, PIASy); SUMO E3 ligase associated with the nuclear matrix could suppress the activity of SMAD3 [118]. Thereby, PIAS4 regulates SMAD-mediated signals by a negative feedback loop. On the other hand, another SUMO E3 ligase, PIAS1 (an inhibitor of activated STAT1), has been identified to promote SMAD4 sumoylation and SMAD-mediated transcription [119].

Table 2 shows the list of inhibitors targeting ERK, PKC (protein kinase C), PKG (protein kinase G), CK1 (casein kinase 1), protein kinase PAK2, PAK4, and LKB1 (liver kinase B1) [120,121,122,123,124,125,126,127,128,129,130,131,132,133,134,135,136,137,138,139]. Others enzyme inhibitors against protein phosphatase, such as (PPM1A/PP2CA, SCP), ubiquitin ligase (NEDD4-2/NEDD4L, HSC70-interacting protein, CHIP, SCFskp1 SCFskp2,) histone acetyltransferase (p300/CREB), and methyltransferase (SET 7/9, SETDB1/ESET, m6 methyltransferase, PRAP1), are listed in Table 2 [140,141,142,143,144,145,146,147,148,149,150,151,152,153,154,155,156,157,158,159,160,161,162,163]. A study by Vena et al. [131] showed that CK1δ is overexpressed in human pancreatic and bladder epithelial cell lines, and the inhibition of CK1δ by application of SR-309 strongly promotes the antiproliferative effect and sensitizes them to gemcitabine treatments. Casein kinase I (CKI) family consists of different isoforms of alpha (α, β, γ1, γ2, γ3, δ, and ϵ), which have been implicated in critical regulatory roles by interacting with R-SMADs. Figure 3 shows that CK1g2 can promote PTMs of SMAD3 activity by deactivation and degradation of the SMAD3 mediated by ubiquitination and phosphorylation [90]. Considering the negative impact of SMAD-mediated signaling, overexpression of CK1δ might dysregulate R-SMAD activity in epithelial cancer cells and activate further SMAD activity to induce SMAD-mediated protumorigenic signals. Upregulation of deoxycytidine kinase (dCK) using the CK1δ inhibitor, SR-3029, may activate the R-SMADs degradation by phosphorylation and promote an antitumorigenic effect. Another potential inhibitor is FRAX597, a small-molecule ATP-competitive Group I PAK inhibitor. It reduces NF2-deficient schwannoma cell proliferation in culture. In vivo, the inhibitor shows potent anti-tumor activity [133]. In NF2-deficient Schwann cells, originating from mesenchymal stem cells, it might inhibit R-SMAD degradation and promote SMAD-mediated protumorigenic signals. It is considered that the application of FRAX597 further deactivates R-SMADs by inducing PTMs, resulting in anti-proliferative and anti-tumorigenic effects in NF2-deficient Schwann cells. Thus, the FRAX597 inhibitor shows significant potential for the treatment of neurofibromatosis and other cancers.

## 5. Regulation of SMAD-Mediated Transcription

### 5.1. Histone Modification

Although SMAD complexes have a low binding affinity, they are guided by many DNA-binding transcription factors, co-factors, repressors, and chromatin modifiers to proficiently control the expression of target genes in a context-dependent manner [164,165,166,167,168,169,170,171]. Here, we discuss and illustrate some SMAD-interacting proteins that regulate SMAD-mediated transcription (Figure 4). It is reported that histone acetyltransferases (HATs) regulate the SMAD-dependent transcription by modifying histones and/or controlling the SMAD activity with chromatin modifiers. Evidence from different studies shows that histone acetyltransferase p300/CBP-associated factor (P/CAF) [172], general control of non-repressed protein 5 (GNC5) [173], switch/sucrose non-fermentable (SWI/SNF) remodeling complex [174], and histone lysine methyltransferase 1 (SETDB1/ESET) [175] are important in executing the function of HATS for SMAD-mediated transcription. The histone acetyltransferases CREB-binding protein (CBP) and p300 govern the SMAD3 transcriptional activity, whereas SMAD4 plays a role as a transcriptional co-activator to stabilize the SMAD complex [176]. P/CAF enhances SMAD3-mediated transactivation driven by the interaction of SMAD3 independently or in association with p300 and [172]. Another HAT, GCN5, plays a role as a co-activator of both TGF-β and BMP-induced SMADs mediated transcription induced by both TGF-β and BMP [173]. SMAD2-mediated transcription is regulated by histone 3 (H3) acetylation with the recruitment of p300 and SWI/SNF and, thereby, confers that chromatin remodeling is necessary for SMAD-mediated transcription [174]. On the other hand, a study by Du et al. [175] shows SMAD-3 mediated recruitment of a histone methaytransferase1, SETDB1/ESET, regulates *snail* gene expression, EMT, and cancer dissemination.

SMAD-driven transcription acts as positive and negative regulators, thereby maintaining the feedback loop of TGF-β/SMAD signaling. The proteins that act as positive regulators are predominantly FOXA10 (Homeobox A10), FOXH1 (Forkhead pioneer factor), G3BO1 (Ras GTPase-activating protein SH3 domain-binding protein 1), AT4 (activating transcription factor 4), SOX4, DLX1, FOXO1-3, E2F4/5, AP-1, CXXC5, and KLF family member proteins (Figure 4).

HOXA10 binds to the SMAD complex and modulates the expression of *snail* and *slug* through m^6^A modification of mRNA and METTL3 expression by enhancing the SMAD signaling [177]. FOX1 can recruit SMAD2 to SMAD4 and assist in binding the SMAD complex with SMAD3:FOXH1 [178]. G3BP1 is another positive co-factor that acts as a novel binding partner to the SMAD complex by activating SMAD signaling and recruiting the SMAD2/SMAD4 complex [179]. The co-factor SOX4 can occupy many genomic loci with SMAD3 in a cell type-specific manner [180]. Forkhead-binding element (FHBE) resides within the SMAD binding region of the p21Cip1 promoter and associates with SMAD2/SMAD3/SMAD4 complex coprecipitated with FOXO1, FOXO2, and FOXO3 [181]. AP-1 and AT4 are downstream regulators of the SMAD complex and mTORC2 and control TGF-β/SMAD and mTOR/RAC1-RHOA pathways independently [182]. DLX1 can positively modulate the TGF-β/SMAD signal by interaction with SMAD4 localized in the nucleus upon TGF-β1 induction [183]. The transcription factor E2F4/5 controls SMAD-mediated transcription in a promoter-specific manner [184,185]. The complex of SMAD3 and the transcription factors E2F4/5, DP1, and corepressor p107 translocate to the nucleus and interact with SMAD4, distinguishing a combined SMAD-E2F site for arresting the cell cycle [185]. Another novel regulator and coordinator of TGF-β, BMP, and Wint signaling is CXXC5, which associates HDAC1 and competes for binding with SMAD complex in hepatocarcinoma cells. CXXC5 reduces the inhibitory effect of HDAC1, resulting in cycle arrest and apoptosis [186,187]. In this context, some Küppel-like factors (KFLs) are well known and can associate with SMADs and act as DNA-binding transcriptional regulators. There are 17 well-known KLFs that have highly conserved C-terminal DNA-binding domains with three C2H2 zinc finger motifs. Some of them play significant roles in feedback regulation in SMAD-mediated transcription. Among them, KLF4, which is up-regulated in vascular smooth muscle cells, recruits p300 and forms an active complex with SMAD2. Thus, KL4 induces the expression of SM22α and α-SMA [188,189]. KLF10 can facilitate multiple TGF-β-induced functions through the expression of SMAD2, inhibition of SMAD7 expression to control the proliferation of epithelial cells, and development of immune and bone cells [190]. Further, KLF11 associates with SMAD3 and enhances TGF-β-induced growth inhibition by repressing SMAD7 and Myc expression in epithelial and pancreatic cancer cells [191].

Among the negative regulators, zinc finger protein OVOL2, 451, histone-binding protein TRIM33 (also E3 ubiquitin ligase activity), HDAC8 (histone deacetylate 8), Ski/SnoN (Sloan Kettering Institute) and SnoN (Ski novel), TGIF, Sp1/KLF-like zinc-finger protein KLF1, and PAX2 are significant to control SMAD-mediated transcription in a cell and context-dependent manner (Figure 4). The zinc finger protein 451 inhibits the recruitment of p300 to the SMAD complex and represses SMAD-mediated transcription, resulting in the reduced H3 Lys9 acetylation of the promoters of target genes [192]. Further, the recruitment of TRIM33 to chromatin is mediated by SMAD4, which promotes histone modification upon binding of TRIM33 and SMAD2/SMAD3 complexes on the regulatory sequences of the target genes. This modification by histone allows the switching of the chromatin state from the poised to the active state and supports the negative feedback mechanism [193].

HDAC8 is class I histone deacetylase, a novel co-factor of the SMAD3/4 complex. A study by Tang et al. [194] shows that chromatin remodeling represses *SIRT7* transcription by making a complex with HDAC8 and SMAD2/SMAD3. In contrast, the reduction of *SIRT7* activates TGF-β by a feedback loop, which regulates the TGF-β/SMAD signal. The negative regulator OVOL2 induces the expression of SMAD7, thereby reducing the expression of SMAD4 and interrupting the complex formation by interfering with the complex formation between SMAD2/SMAD3 and SMAD4 [195]. Also, the negative regulators Ski and SnoN interact simultaneously with the R-SMADs and SMAD4 and disrupt the ability of the SMAD complexes to turn on the target genes [196,197]. Ski/SnoN actively recruits a transcriptional co-repressor complex containing N-CoR/HDAC to the targeted promoter by preventing the recruitment and binding of R-SMADs to p300/CBP. It has been shown that 5′TG3′-interacting factor (TGIF) represses TGF-β signaling [198] by binding with R-SMAD/SMAD4 complexes. Further, Guca et al. [199] showed that TGIFI-HD (homeodomain) binds to SMADs in a mutually exclusive manner. Thereby, it provides a transcriptional repression system in a context-dependent manner. Among the KLF family, KLF2 employs a negative feedback loop on TGF-β/SMAD signaling by inducing SMAD7 transcription [200]. A negative feedback loop is maintained to regulate TGF-β/SMAD signaling by the action of I-SMADs. It is found that in the absence of a ligand, TGF-β promotes nuclear accumulation of I-SMADs from the nucleus to cytoplasm and promotes SMAD7 mRNA and, thereby, creates a negative feedback loop that firmly controls TGF-β SMAD signaling [201,202]. Enhanced SMAD7 methylation by SET9-mediated TGF-β/SMAD signaling promotes its association with the E3 ubiquitin ligase Arkadia and enhances its ubiquitylation and degradation in lung fibroblast. Pharmacological inhibition or depletion of Set9 showed that high SMAD7 protein levels inhibited the expression of TGF-β/SMAD-mediated genes encoding extracellular matrix components [203].

### 5.2. Regulation of SMAD-Mediated Transcriptional Activity Post-Transcriptionally

At the post-transcriptional level, SMAD-mediated transcription is regulated by RNA-binding proteins (RBPs), microRNA (miRNA), and non-coding RNAs (ncRNAs). A study by Tripathi et al. [204] shows that the interaction of phosphorylated T179 of SMAD3a with PCBP1 (RNA-binding protein) promotes alternative cancer stem cell marker CD44 splicing. SMAD3 and PCBP1 can cooperate in the variable regions of exons for CD44 pre-mRNA and modulate spliceosome assembly. As a result, mesenchymal isoform CD44s is expressed over epithelial isoform CD44s. Thus, alternative splicing by SMAD3 plays an important role in tumorigenesis. Interestingly, R-SMADs are found to influence miRNA expression, processing, and maturation of 44 types of miRNAs. R-SMADs can bind to the stem region of miRNA at a consensus sequence and recruit p68 (DDX5), an RNA helicase component, DROSHA (DROSHA/DGCR8/RNA) in a ligand-dependent manner. Thus, the post-transcriptional modification induced by TGF-β and BMP signaling can drive increased expression of mature miRNA-21 after processing of primary transcripts of miRNA-21 (pri-miRNA-21) into precursor miRNA-21 (pre-miRNA-21) [205,206]. In addition to miRNA-21, other miRNAs are directly involved in the feedback regulation of SMAD-mediated signaling, as reported by Yan et al. [207]. Additionally, long-coding RNAs (lncRNAs) are recognized as intermediaries of the TGF-β response. A comprehensive analysis by Adylova et al. [208] shows that the different lncRNAs can play a role in positive and negative regulation of TGF-β/SMAD signaling in different cancer cells. Another post-transcriptional negative feedback has been described by Bertero et al. [209] in human pluripotent stem cells in which R-SMADs can associate with the m^6^A methyltransferase complex METTL3-METTL4-WTAP. In the nascent transcripts, N^6^-adenosine methylation on the RNA occurs by the action of the m^6^A methyltransferase, resulting in destabilizing and degrading the transcripts [209].

Table 3 shows the list of inhibitors targeting histone acetyltransferases (GNC5/PCAF, SWI/SNF, histone lysine methyltransferase (SETDB1/ESET), small GTPases (RAS), histone deacetylases (HDAC1, HDAC8), and methyltransferase (m^6^A methyltransferase, SET (7/9) methyltransferase) [123,160,210,211,212,213,214,215,216,217,218,219,220,221,222,223,224,225,226,227,228,229,230,231,232,233,234]. Some of the potential inhibitors are highlighted here. The application of microRNAs (miRNAs) for cancer therapy and radiosensitivity in tumors has achieved significant consideration. A study by Shao et al. [217] showed that miRNA-621 could enhance the radiosensitivity of hepatocellular carcinoma (HCC) through direct inhibition of SETDB1 and targeting the 3′UTR of SETDB1. It was reported that miRNA-621 and/or the SETDB1 axis activated the p53-signaling pathway and advanced the radiosensitivity of HCC cells. SETDB1/ESET regulates SMAD-mediated transcription to control *snail* gene expression in epithelial or mesenchymal cells in a context-dependent manner. It is supported that *snail* may bind directly to the DNA-binding domain of p53 and reduce the p53 tumor-suppressive function. Since the expression of miRNA-621 and SETDB1 are negatively correlated in HCC tissues, miRNA-621 might enhance the radiosensitivity and active p53 signaling pathway in HCC cells by inhibiting SETDB1 expression. Thus, mi-RNA-621 and/or SETDB1 in epithelial cells of HCC can be potentially used as a novel therapeutic target [217].

STM2457 is the first bioavailable small-molecule inhibitor of METTL3 (a regulator of m^6^A methyltransferase) that affects the inhibition of catalytic activity and upregulation of METTL3 increasing PD-L1 and reduction of tumor progression in NSCLC (non-small-cell lung cancer) [233]. Upregulation of METTL3 induced by STM2457 enhances the interaction of METTL3 with the translation initiation machinery to make more circularized mRNA of PDL-1. In this case, STM2457 may inhibit R-SMADs from associating with the m^6^A methyltransferase complex METTL3 to destabilize and degrade the nascent transcripts of PDL-1. Thus, STM2457 is considered a potential suppressor that provides an inhibitory effect in epithelial cells of NSCL.

The SET domain containing lysine methyltransferase (SETD7/9) is involved in various disease-related signaling pathways with a broad group of substrates. Methyltransferase activity of human SETD7 is inhibited by a selective inhibitor, (*R*)-PFI-2. It was found that the Hippo pathway was modulated by (*R*)-PFI-2 with increasing nuclear YAP and YAP-mediated gene expression in epithelial cancer cells (MCF7) and murine embryonic fibroblasts [234]. How (*R*)-PFI-2 works is not well defined, but a crosstalk between TGF-β/SMAD and Hippo signaling suggests that the binding of YAP to SMAD7 may modulate TGF-β signaling. SMAD7 methylation by SET (7/9)-mediated TGF-β/SMAD signaling might promote its association with E3 ubiquitin ligase, thereby enhancing its ubiquitylation and degradation. YAP can associate with the complexes of R-SMADs-SMAD4 and drive their sub-cellular localization and transcription in a context-dependent manner. Thus, (*R*)-PFI-2 and related compounds can be valuable inhibitors to target methyltransferases.

## 6. Non-SMAD, Non-Canonical TGF-β Pathway Control

### 6.1. ERK/MAP Kinase Signaling

The Ras/Raf/MAPK (MEK)/ERK pathway is one of the important signaling cascades and contributes an important role in tumor cell survival and progression. TGF-β activates the ERK-MAP kinase pathway by direct phosphorylation of ShcA, which subsequently activates downstream signals (Figure 5A). A study in both Mv1Lu mink lung epithelial cells and mouse fibroblasts by Lee et al. [235] showed that phosphorylation of ShcA on tyrosine results in the initiation of a docking site for the downstream mediators Grb2 and Sos, which further activate Ras GTPase, Raf, MEK (MAP kinase/ERK kinase), and EK1/2 kinase. In conjugation with SMADs, Erk regulates the transcription of target genes through the downstream transcription factors and regulates EMT. ERK activation in human keratinocytes (HaCaT) is initiated by the TGF-β receptor complexes, which are localized in lipid rafts. Clathrin-dependent endocytosis of TGF-β receptor complexes initiates SMAD activation, SMAD-mediated transcription, and TGF-β/SMAD-directed epithelial cell plasticity [236].

### 6.2. JNK and p38 MAP Kinase Signaling

Jun N-terminal kinases (JNKs) and p38 mitogen-activated protein kinases (MAPKs) significantly coordinate with many signaling mechanisms associated with stresses. In addition, different cellular functions are regulated by the JNK and p38 pathways. In brief, TRAF6 (Tumor Necrosis Factor Associated Receptor-Associated Factor 6) interacts with the TGF-β receptor complex once receptors are activated by ligand binding. An autoubiquitylation modification induces TRAF6, which activates TAKI by polyubiquitylation at Lys^63^TAK1 and initiates the p38 MAPK pathway (Figure 5B) [237,238]. Importantly, TAK1 is a negative regulator of canonical TGF-β signaling and promotes R-SMAD phosphorylation at the linker region in neural crest-derived mesenchymal cells [239]. In this pathway, I-SMADs play significant roles in regulating p38 MAP kinase signaling. It was found that SMAD7 in prostate cancer cells can act as a scaffolding protein for p38 and its upstream kinases [240]. Also, in AML-12 mouse liver cells and primary hepatocytes, SMAD6 acts as a negative regulator and eliminates K63-linked polyubiquitylation of TRAF6 by recruiting the A20 DUB enzyme induced by TGF-β1 [241].

Another ligase, E3, and TRAF4 activate the MAP kinase pathway (Figure 5C). In this pathway, TRAF4 is auto-ubiquitylated upon ligand binding and is recruited to the TGF-β receptor complex. In breast cancer cells, TRAF4 activates TAKI via polyubiquitylation, which results in the activation of the p38 pathway. Subsequently, SMURF2 is degraded through polyubiquitylation by TRAF4, thereby maintaining the stability of the TGF-β1 receptor [242].

### 6.3. JAK-STAT Signaling

JAK/STAT (the Janus Kinase Signal Transducer and Activator of Transcription) pathway is a prime regulator of cell function. (Figure 5D). Jak/STAT-mediated downstream effects may vary and drive hematopoiesis, tissue repair, inflammation, immune surveillance, apoptosis, and adipogenesis [243]. A study by Tang et al. [244] shows that JAK1 is a constitutive TGF-βRI that is essentially involved in STAT phosphorylation within a short period after TGF-β stimulation. Once SMADs are activated, another phosphorylation of STAT is started for de novo protein synthesis. Thus, the non-SMAD JAK1/STAT pathway in hepatic stellate cells is essential for the expression of TGF-β subset genes.

### 6.4. PI3/AKT/mTOR Signaling

The two pathways, PI3K (Phosphatidylinositol-3-kinsae)/Akt and Mammalian Target of Rapamycin (mTOR), play roles in many different cellular functions (Figure 5E). In epithelial cells, the TGF-β type I receptor (TGF-βRI) can bind PI3K and modulate the kinase activity [245]. Further, a study by Hamidi et al. [246] demonstrated that in prostate cancer cells, TRAF6 can polyubiquitylate p85α, a regulatory subunit of PI3K, and make a complex between p85α and TGF-βRI to activate PI3K and AKT. Moreover, Lamouille et al. [247,248] showed that TGF-β can induce mTORC2 (Mammalian Target of Rapamycin Complex 2), which further phosphorylates and activates AKT, resulting in cell size changes and epithelial cell progression through EMT.

### 6.5. TGF-β Type I Receptor (TGF-βRI) Intracellular Domain Signaling

TGF-βRI intracellular domain signaling (Figure 5F) promotes TGF-β-mediated tumor invasion. It is reported by Mu et al. [249] that in prostate cancer cells, TGF-β uses TRAF6, resulting in Lys63-linked polyubiquitination of TGF-βRI and promoting cleavage of the extracellular domain of TGF-βRI by TACE (TNF-alpha converting enzyme). The newly formed intracellular domain (ICD) of TGF-βRI can bind with p300 for transcription of genes, thereby driving tumor invasion by induction of SNAIL and MM2 [250]. TRAF6 can polyubiquitylate a membrane-bound protein, PS-1 (membrane-embedded protease presenilin-1), to initiate a proteolytic cleavage on TGF-βRI. Thereby, the ICD (the complete intracellular domain) of the receptor is formed to assist the translocation of TGF-βRI-ICD to the nucleus [251]. In the nucleus, TGF-βRI-ICD binds to the promoter and turns on the transcription of the genes encoding TGF-βRI. In prostate cancer, TRAF-6-mediated Lys6-linked ubiquitination of the TGF-βRI intracellular domain is important for the modulation of TGF-β and regulation of other genes controlling the cell cycle, proliferation, differentiation, and migration [251].

### 6.6. Rho-(like) GTPase Signaling

Rho GTPases encompass a branch of the Ras superfamily with 22 genes in humans, and among them, RhoA, Rac1, and Cdc42 are the best exemplified (Figure 5G). They have a significant role in many cellular processes, mainly in cell morphology regulation, cell adhesion, and cytokinesis production. In many studies of mammalian cells, the constitutive and dominant negative mutants were used to examine the function of Rho GTPases. Bhowmik et al. [252] showed that in epithelial cells, the RhoA-dependent signaling pathway stimulates the formation of stress fibers induced by TGF-β. As a result, epithelial cells can transform into cells with mesenchymal characteristics such as increased N-cadherin expression and motility with the loss of E-cadherin (markers of TGF-β-induced EMT). TGF-β induction can be reduced by blocking RhoA or its downstream target, p160^Rock^, expressed by the dominant-negative mutants. Another study by Edlund et al. [253] showed that TGF-β1-treated human prostate carcinoma cells (PC-3U) can promptly form lamellipodia by rearranging the actin filament system. This response was independent of SMAD for a short time, but it requires Rho GTPases Cdc42 signaling for the long term.

A regulator of epithelial cell polarity, Par-6, can negatively control Rho-GTPase signaling by cooperating with TGF-β [254]. Par6 is associated with the TGF-βRI, phosphorylated by TGF-βRII, and thereby it recruits SMURF1 ubiquitin ligase (Figure 5G). Rapid degradation of RhoA GTPase is initiated through polyubiquitylation by SMURF1. This introduces the loss of a tight junction of epithelial cells. Further, Wilkes et al. [255] showed that signals from the TGF-β receptor activate PAK2 (STE20 homolog) in mammalian cells. PAK2 activation was observed in fibroblasts (not in epithelial cell cultures) mediated by signals from SMAD2 and/or SMAD3. In fibroblasts, Rac1 and Cdc42 might regulate PAK2 activity induced by TGF-β. However, targeting PAK2 by morpholino antisense oligonucleotides and dominant negative PAK2 can prevent the morphological features. Thus, PAK2 is considered a novel SMAD-independent pathway that distinguishes TGF-β signaling in fibroblasts and epithelial cells.

### 6.7. Crosstalk between SMAD and Other Signaling Pathway Molecules

The intracellular signaling network, the SMAD pathway, and the other pathways are connected to develop crosstalk. The crosstalk within the pathways plays an important role in the regulation of biological processes. The crosstalk can occur at different levels by altering signaling components, transcriptional modification, and chromatin modification or by direct interactions between intracellular signaling components with SMADs. Here, we briefly mention the crosstalk of SMADS with Yes-associated proteins with a PDZ-binding motif (YAP/TAZ) and interactions with other proteins, such as TRAP-1, km23-1, and PKA. The binding of YAP to SMAD7 by TGF-β and Hippo signaling was the first reported crosstalk that resulted in the increased inhibition of TGF-β signaling [256]. YAP can bind to the PpxY motif and phosphorylate the SMAD1 linker region by CDK9 in mammalian cells [257]. Neural cell (mESC) differentiation by BMP signal is suppressed by YAP-SMAD1 binding and further SMAD-1-dependent transcription. TAZ and YAP can associate with a heteromeric complex of R-SMAD-SMAD4 mediated by TGF-β signaling [258]. In contrast, another study showed that SMAD nuclear localization in response to TGF-β is not dependent on the levels of YAP or TAZ [259]. Also, it was found that the localization of YAP/TAZ can overcome cell cycle arrest stimulated by TGF-β and promote a pro-tumorigenic transcriptional program [260]. It is reported that TGF-β/SMAD/YAP/TAZ crosstalk also plays an important role in cell differentiation, proliferation, and fibrogenesis in a context-dependent manner [259,260].

The protein kinase A (PKA) signaling pathway is described by Wang et al. [261] in mesangial cells. PKA activates TGF-β stimulated cAMP response element-binding protein phosphorylation and expression of fibronectin. A study by Yang et al. [262] reported that PKA is independent of cAMP by aiding an interaction of PKA holoenzyme subunits with activated SMAD2/SMAD3. The interaction domains of SMAD4 and PKA-R and their functional roles are defined by the study. It was shown that amino acids 290–300 of the SMAD4 linker region are critical for the specific interaction of SMAD4 and PKA-R for the regulation of TGF-β-mediated cellular functions (e.g., PKA activity, CREB phosphorylation, induction of p21, and inhibition of growth). In pancreatic cancer cells, SMAD-PKA plays a role in TGF-β-induced EMT and tumor growth.

TLP is a TRAP-1-like protein that acts as an adaptor protein that mediates the interaction with the TGF-β type II receptor and the downstream effector SMADs. Felici et al. [263] proposed through a study that SMAD2 and SMAD3 might be balanced by TLP signaling through the localization of SMAD4 intracellularly. Thus, the specificity of TGF-β transcriptional responses can occur in a context-dependent manner.

Table 4 shows the inhibitors of SMAD-interacting enzymes (Jun N-terminal kinase (JNK), p38 MAP kinase, AKT, TRAF6, RhoA, PKA) that control non-SMAD noncanonical TGF-β pathways [264,265,266,267,268,269,270,271,272,273,274,275,276,277,278,279,280,281,282,283,284,285,286,287,288]. A potential inhibitor, SP0016125, targets c-Jun N-terminal kinase (JNK) and stimulates TGF-β-induced apoptosis of the RBE human cholangiocarcinoma cell line [264]. The result showed that SP0016125 increased the TGF-β-induced SMAD2 and SMAD3 phosphorylation, which promoted TGF-β1-induced transcriptional response and apoptosis in RBE cells. Depletion of SMAD4 reduced the effect of SP600125 on the transcriptional response and apoptosis. Further, TGF-β-induced apoptosis was abolished using the pan-caspase inhibitor Z-VAD-fmk. These findings indicate that SP600125 enhances TGF-β-induced apoptosis of RBE cells by promoting SMAD-dependent caspase activation through a SMAD-dependent pathway. Further, a study by Lu et al. [265] showed a novel pro-apoptotic role in combination with dihydroartemisinin (DHA). In human lung adenocarcinoma cells (ASTC-a-1) induced by DHA, SP600125 synergistically enhances apoptosis through Bax translocation and other intrinsic apoptotic pathways like caspase activation and the mitochondrial pathway. These findings suggest that SP600125 may enhance TGF-β-induced apoptosis in lung adenocarcinoma through a SMAD-dependent caspase pathway. Therefore, SP0016125 is considered a novel therapy for the treatment of cholangiocarcinoma and lung adenocarcinoma epithelial or mesenchymal cells that express JNK.

A potential inhibitor named LY2228820 (an imidazole derivative) is a selective ATP-competitive inhibitor of the α-and β-isoforms of P38α mitogen-activated protein kinase (p38 MAPK) [274]. In vivo cancer models of breast, ovarian, lung, glioma, and myeloma showed that tumor growth was potentially delayed with LY2228820 treatment. Since p38α regulates cytokine (TNF, IL-Iβ, IL-6, IL-8, etc.) production in TME, it is considered that TRAF6 expression in multiple cancer cell lines (MDA-MB-468, OPM-2, A549, A2780) may also control cytokine production. Further, it is hypothesized that auto-ubiquitylation of TRAF6 upon ligand binding might result in recruiting the TGF-β receptor complex, which in turn actives TGF-β/SMAD-mediated target genes to stimulate EMT. Therefore, LY2228820, a p38 MAPK inhibitor, has been optimized for various purposes such as potency, selectivity, bioavailability, and efficacy in animal models of human cancer.

Another important inhibitor of the non-SMAD, noncanonical TGF-β pathway is PKI protein targeting PKA, a well-known regulator of physiological and oncogenic functions. A study by Hoy et al. [289] found that PKI can modulate tumor growth by a molecular switch to drive GPCR-Gαs-CAMP signaling toward EPAC-RAP1 and MAPK. Amplification of *PKIA* is common in prostate cancer cells, and depletion of *PIKA* showed reduced migration and tumor growth. To understand the mechanism of PKIA, it is hypothesized that the expression of PIKA and EPAC may modulate MAPK or ERK/MAPK signaling. In conjugation of SMADs, Erk may regulate downstream gene expression for EMT in prostate cancer cells.

## 7. TGF-β/SMAD Mediated Progression in Solid Tumors

The TGF-β/SMAD signaling pathway plays a dual role in cancer progression. In addition to promoting apoptosis and cell cycle arrest in epithelial cells, TGF-β/SMAD and non-SMAD signaling enhances EMT, cell migration, invasion, angiogenesis, and stemness. TGF-β/SMAD exhibits a tumor suppressor phenotype in epithelial cells at the early stages of tumorigenesis. In contrast, at the later stages, TGF-β/SMAD signals drive oncogenesis and metastasis in mesenchymal properties of cancer cells (Figure 6). Here, we show SMAD-dependent canonical and non-canonical signals mediated by cancer cells with epithelial, EMT, hybrid/partial EMT, and mesenchymal properties associated with cancer progression and metastasis.

### 7.1. Epithelial Cells

The typical characteristics of epithelial cells are described and summarized by Huang et al. [289]. Briefly, epithelial cells comprise strong apical and basal polarity with plasma membranes placed toward and away from the lumen, respectively. Different proteins are found in each membrane. Proteins in the membrane assist in the transportation and localization of targeted molecules to specific cellular regions with various activities (Figure 6). Both SMAD-dependent canonical and non-canonical pathways predominantly regulate non-cancerous epithelial cells at the pre-malignant stage by cell cycle progression through CDK (G1-Arrest Activating Cyclin-Dependent Kinase) inhibitors p15 and p21. Further, the downregulation of an important oncogene, c-Myc, drives the proliferation and inhibits the transcriptional activity of p15 and p21 [3]. Importantly, apoptosis is initiated by the signals modulating the expression of B-cell lymphoma-2 (BCL-2) family members, BIM (BCL2L11), FAS (death receptor fibroblast-associated antigen (FAS)), DAPK (death-associated kinase), BH3-protein BIK, and caspases, which induce both intrinsic and extrinsic apoptosis [290,291,292]. Further, tumor suppression mediated by TGF-β/SMAD signaling has been demonstrated in some solid tumors like breast cancer, prostate cancer, melanoma, and colon cancer [293]. The non-canonical TGF-β/SMAD is linked to p38 MAPK and caspase-8-dependent programmed cell death, which exhibits a tumor suppressor role by inducing apoptosis [294]. Further, to drive tumor cell death by activating programmed cell death, TGF-β/SMAD promotes the regulation of immune cell function [295,296].

### 7.2. Cancer Cells with EMT, and Hybrid/Partial EMT

The transition of cancer cells is dynamic, allowing cells to go from epithelial to mesenchymal states and vice versa. A change from epithelium to the mesenchymal state is referred to as EMT (epithelial–mesenchymal transition), which allows cells to gain migratory properties by modifying the adhesion molecules expressed by the cells. The reverse process is MET (mesenchymal–epithelial transition), which is associated with loss of migratory properties rather than adopting an apicobasal polarization [4]. Thus, beyond epithelial phenotype, various EMTs can impact cell behaviors, physiology, and ecology, allowing the transition of different phenotypes (Figure 6). The EMT is implemented for pleiotropic signaling through canonical and non-canonical SMAD pathways. Thereby, specific transcription factors (TFs) named EMT-TFs (e.g., SNAIL, ZEB, TWIST) are expressed to regulate EMT and MET.

Along with transcription, miRNA, post-translational and epigenetic regulators mediate EMT in cancer progression [289]. EMT has been established as a spectrum rather than a linear process supported by recent studies [5,6,7] (Figure 6). A study by Pastushenko et al. [6] used different markers, CD106, CD51, and CD61, showing that the different states of the EMT spectrum can form a “hybrid” (Figure 6). A single-cell analysis found that partial/hybrid EMT expresses SNAIL1/2, ZEB1/2, and TWIST1/2 during mouse organogenesis in a manner similar to epithelial gene expression [297]. Further, using the single-RNA sequencing technique with primary and metastatic head and neck squamous cells, it was shown that that cell contained a gene signature of partial EMT [298]. Partial EMT is also highly aggressive and invasive since the cells present several classical features of EMT. Expression of vimentin (VIM), TGF-β1, and extracellular matrix genes are worth mentioning. Although the EMT-TF expression was low, transcription of epithelial genes was still maintained within the partial EMT. The EMT phenotype can be variable, such as pancreatic ductal adenocarcinoma resulting from adjusting epithelial junction proteins that showed no change in expression, contrasting the common repression of epithelial characteristics of cells [299]. Moreover, many carcinoma cells, such as breast and colorectal cancer cells, can utilize this partial/hybrid EMT program to make a cluster of cells contrasting the single-cell pattern, which is connected to traditionally defined EMT mechanisms [299].

In addition, EndMT (endothelial–mesenchymal transition) shows similarity to EMT in the context of molecular processes and phenotypes. The loss of endothelial junctions, EMT-TF activation, and upregulation of mesenchymal markers push EndMT formation. Endothelial cells downregulate the VE (vascular endothelial cadherin), where cell-type-specific changes distinguish EndMT from EMT [300].

### 7.3. Cancer Cells with Mesenchymal Characteristics

Upon EMT, cells repress their epithelial phenotypes and gain mesenchymal and invasive properties during cancer progression. The hallmarks of EMT are increased N-cadherin and VIM expression with decreased expression of E-cadherin, ZO1, and desmoplakin [301,302,303]. In cancer cells with mesenchyme morphology, the major EMT-TF are zinc finger binding transcription factors, including SNAIL1/2, E-box binding homology frame factors (ZEB1/2), and BHLH (basic Helix–Loop–Helix) factors (TWIST1/2) [304]. Both SNAIL1 and SNAIL2 drive tumorigenesis induced by EMT. During the upregulation of mesenchymal phenotypic markers, including VIM and fibronectin, SNAIL1 directly inhibits TJ formation by decreasing epithelial markers like E-cadherin and claudin expression. E-cadherin positively correlates to patient survival, whereas the overexpression of MMPs provides aggressiveness of tumors [305]. On the other hand, SNAIL2 promotes the loss of cell adhesion and polarity by decreasing e-calmodulin levels, thereby influencing migration and metastasis in breast and ovarian cancer [306,307,308].

In tumor stem cells, ZEB1/2 expression in epithelial cells causes EMT and mesenchymal phenotypes with invasiveness, metastatic, and dedifferentiation potential [309]. In vivo study of pancreatic cancer supports that ZEB1 is essential for tumor invasion and metastasis [310]. Collective data show that ZEB1/2 expression in breast, colorectal, and pancreatic cancers affects poor patient outcomes [311,312,313,314].

Like SNAIL, TWIST1 can suppress the expression of E-cadherin and promote N-cadherin expression, which affects cell adhesion and cell motility [315,316]. Tumor invasion and metastasis are driven by the overexpression of TWIST [317]. TWIST can be activated by various signals, such as HIF-1α (hypoxia-inducible factor-1α), during the progression of EMT. In a hypoxic condition, activation of TWIST by HIF-1α allows the cell to disseminate with metastatic potential [315]. Further, in a mouse model of squamous cell carcinoma, it was found that TWIST can facilitate the cancer cells to undergo EMT and dissemination in the circulation [318].

### 7.4. Cancer Cells with Stemness Characteristics

Cancer cells have a group of tumor-initiating cells known as cancer stem cells (CSCs), which can differentiate, renew, and generate tumor heterogeneity within the populations of cells [319]. CSCs studied in breast cancer models revealed that EMT correlates with CSC generation by repressing epithelial and mesenchymal properties [320]. It was reported that the expression of CSC and EMT markers were observed in mammary epithelial and carcinoma cells by the direct induction of TGF-β/SMAD signaling, which promotes mammosphere, soft agar colony, and tumor formation [321]. Also, it was shown that autocrine TGF-β/SMAD signaling is crucial within the subpopulation of immortalized breast epithelial cells to maintain its mesenchymal phenotype and tumorigenicity [321]. It is suggested that in this system, BMP signaling may or may not antagonize TGF-β-induced EMT and CSC generation [322]. Thus, enhanced TGF-β/SMAD signaling can promote EMT and CSC properties in cancer cells, further allowing CSC in cancer cell invasion and dissemination.

### 7.5. Cancer Dissemination and Metastasis

Tumor cells interact with ECM components (collagen, fibronectin, laminin) and cells in the tumor stroma in vivo. In TME, TGF-β/SMAD plays significant roles in cancer cells driven by autonomous tumor cell signaling. Cancer metastasis is promoted in stromal fibroblast and mesenchymal stem cells stimulated by TGF-β/SMAD signaling [8,9,10,323]. Therefore, ECM and stroma are the primary sites for cancer dissemination and metastasis, which are initiated through invasion. Further, intravasation into the blood circulation, extravasation at distant sites, and adaptation at a secondary site within a new microenvironment proceed [324].

Epithelial plasticity plays a vital role in invasion by carcinoma cells and further promotes their dissemination and metastasis [325,326]. Mesenchymal gene expression is often increased in circulating tumor cells, which disseminate through individual as well as collective cell migration [327]. However, many carcinoma cells, such as breast and colorectal cancer cells, can utilize partial/hybrid EMT programs that favor migration as clusters over signal cell migration defined by traditional EMT migration [301]. Interestingly, leading cells in a cluster with mesenchymal characteristics boost invasion, whereas most stalking cells are hybrid/partial EMT or remain epithelial [328]. At the invasive edges, cells with EMT plasticity respond to TGF-β/SMAD signaling to facilitate dissemination through blood vessels or lymphatics [326]. In squamous cell carcinoma, SNAIL induces the expression of claudin-11, a tight junction protein, and advocates cell migration as a tumor cell cluster [278].

Reversible EMT is supported by a model in which transient expression of EMT transcription factors affects the reversion of EMT and metastatic colonization [11,12,13]. TGF-β/SMAD signaling, which stimulates EMT and dissemination, needs to be repressed for metastatic reversal to epithelial morphology [323]. Breast cancer cells can switch from cohesive to single-cell motility mediated by localized and reversible TGF-β/SMAD. To examine SMAD localization in breast cancer progression, Giampieri et al. [323] demonstrated that TGF-β is activated locally and transiently in a motile cell. A transcriptional program involving several factors (SMAD4, NEED9, EGFR, RhoC) can actively modulate cell motility from cohesive to a single cell. Inhibition of signaling can prevent single-cell motility, but collective cell migration was not inhibited. However, TGF-β/SMAD signals can stabilize mesenchymal phenotypes within the tumor cells and are not shown as supportive for leading MET.

A study by Katsuno et al. [329] showed that the prolonged TGF-β/SMAD signals could promote EMT in epithelial cancer cells. Short exposure to TGF-β/SMAD induces the opposite effect and drives MET for colonization in the lung. Moreover, it was shown that the prolonged TGF-β/SMAD signals further modulate mTOR signaling, which contributes to cancer cell stemness and drug resistance. Therefore, it is important to either short exposure of TGF-β/SMAD signals or other signals to balance between TGF-β signaling and switching of EMT to MET. Another study showed that EMT reversed during migration, not in circulation. This study observed MET that was formed after enormous cycles of cell division to reach a secondary site during colonization [11]. Furthermore, it was found that the hybrid EMT has high tumor-initiating and self-renewal capacity in primary cancer cells, especially in breast and prostate cancer [330,331]. Therefore, it is suggested that EMT plasticity and stemness can play a role in the metastatic colonization of tumor cells. Future studies are required to understand the mechanism of EMT plasticity in cancer cell progression, dissemination, and metastasis.

Dormancy is another stage of the tumor cells that allows the cell to be dormant in an arrest phase at primary or secondary sites [332]. A recent review by Fares et al. [331] described that a delayed adaption of disseminating cancer cells (either single invading or cluster cells in circulation) to their secondary at a new microenvironment causes dormancy. TGF-β/SMAD signals induce the tumor suppressor gene DEC2 (chondrocyte 2), which assists cells in entering a quiescence state by inhibiting CDK4 and activating p27.

Tumor dormancy is maintained by the intracellular signals from the two important SMAD-independent pathways, RAS-MEK-ERK/MAK and PI3K-AKT [333].

Some studies [334,335] show that EMT and cancer cell dissemination can occur at the pre-malignant or epithelial stage of tumorigenesis, contrasting with the concept of whether mesenchymal cells aid metastasis in the later stage of tumorigenesis. Before primary tumors are detected, invasive cells with EMT characteristics are detected in models of various solid tumors, such as in pancreas, lung, and breast cancer cells. The evidence supports that such cells can disseminate and then undergo dormancy. Further reactivation of dormancy allows them to form metastatic tumors

## 8. Conclusions

This review highlights the comprehensive understanding of the functions of SMADs and SMAD-interacting and/or SMAD-signal-modulating proteins in tumorigenesis. The phenotypes of solid tumors are flexible and can switch through epithelial, EMT, partial/hybrid EMT, mesenchymal, and MET states. This phenotypic plasticity creates challenges to developing new therapies for the treatment of cancer. Therefore, we summarize the important enzymes (involving both TGF-β/SMAD and non-SMAD-mediated TGF-β signaling pathways) and the inhibitors targeting the enzymes (Table 1, Table 2, Table 3 and Table 4). The preclinical findings of the inhibitors on solid tumor models have the potential to understand their inhibitory mechanisms and future perspective. The inhibitors can act by different mechanisms that induce cell cycle arrest, apoptosis, modulation of EMT, and dissemination, thereby inhibiting cancer cell proliferation and tumor growth. Although the findings are preliminary, some considerations are required for prospects. Detection of novel therapies is essential to increase efficacy. Further well-designed clinical trials are important to validate safety, effectiveness, and tolerability with the patients.

## Figures and Tables

**Figure 1 pharmaceuticals-17-00326-f001:**
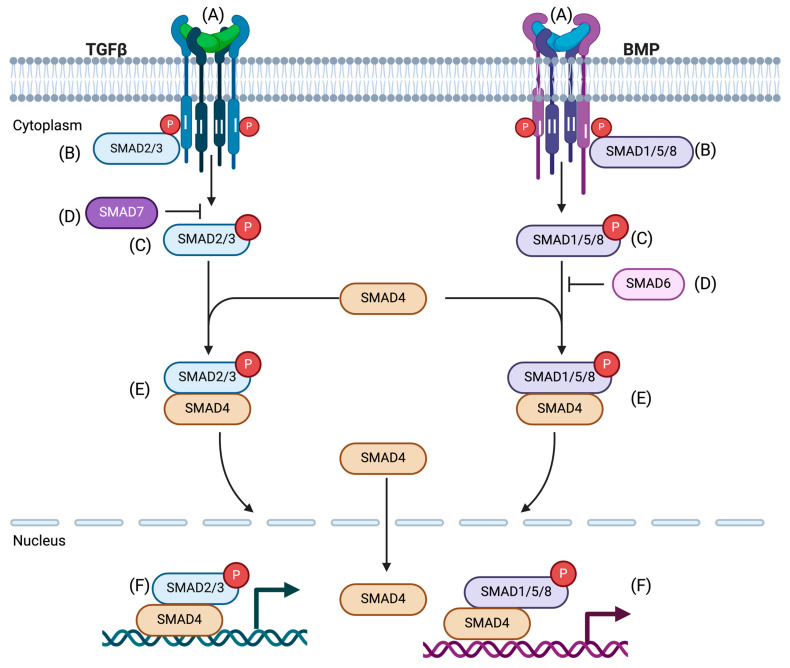
**The TGF-β/SMAD signaling pathway.** (**A**) Ligand binding: A heteromeric complex is formed with an interaction of type II, type I receptors, and TGF-β ligands. (**B**) Activation of receptor: Ligand binding stimulates the type II receptor to phosphorylate and activate the type I receptor. (**C**) Activation of R-SMADs: Once the type I receptor is activated, it initiates phosphorylation and activation of the receptor-activated SMADs (R-SMADs, SMAD2, and SMAD3 for TGF-β signal). (**D**) Inhibition of R-SMAD activation: Inhibitory SMADs, SMAD7, and SMAD6 compete with R-SMADs to intervene with the type I receptor. As a result, R-SMAD activation and transmission of the SMAD signaling is prevented. (**E**) The complex of R-SMADs and CO-SMAD formation: Activated R-SMADs dissociate from type I receptors to form a complex with the common mediator, CO-SMAD, SMAD4. (**F**) The complex of R-SMAD and SMAD4 can translocate from cytoplasm to nucleus and bind to transcription factors, resulting in the transcription of target genes. Here, SMAD1, SMAD5, and SMAD8 are shown for BMP signal. Adapted from *Molecular Biology of Human Cancers* (Springer 2005) [28].

**Figure 2 pharmaceuticals-17-00326-f002:**
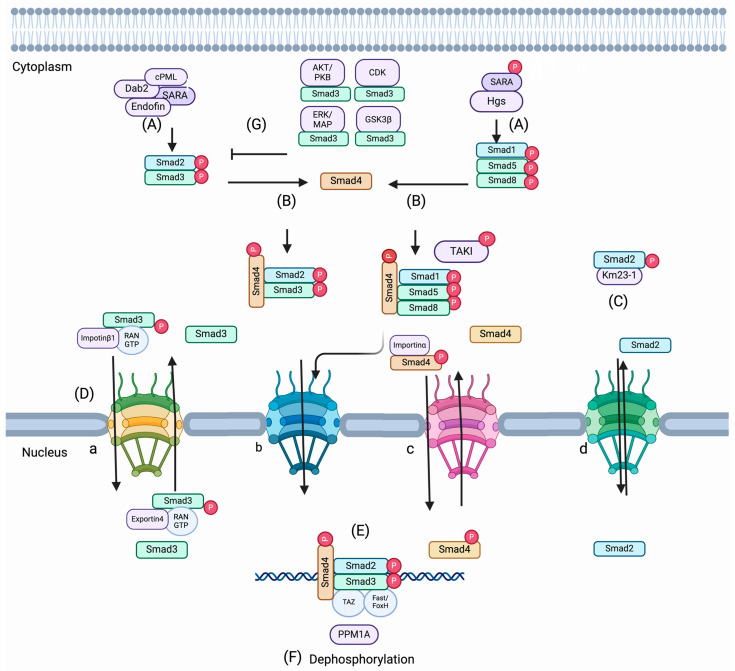
**Activation and nucleocytoplasmic trafficking of SMADs.** (**A**) The association of endocytic protein SARA (SMAD Anchor of Receptor Activation) and a few SARA-associated proteins, cPML, endofin, and Dab2, promote R-SMAD activation R-SMADs to the TGF-β receptor. Hgs (hepatocyte growth factor-regulated tyrosine kinase substrate) is associated with SARA and promotes phosphorylation of SMADs 1, 5, and 8 and TAKI induced by BMP signal (**B**). Once R-SMADs are phosphorylated and dissociated from the receptor complex, SMAD4 and phosphorylated R-SMADs can interact and make a trimeric complex. (**C**) A dynein light chain km23-1 interacts with SMAD2 and SMAD3 and assists in nuclear translocation. (**D**) Nuclear translocation of SMAD complexes is maintained by nuclear pore proteins. Importin/exportin family members and Ran proteins (small Ran GTPase) are also involved in the trafficking of R-SMADS and SMAD4. (**a**) Importin/exportin-mediated Ran-dependent method; (**b**) Phospho-dependent method; (**c**) Importin-mediated Ran-independent method; (**d**) Importin/exportin-independent and phospho-independent method. (**E**) SMADs can be reserved in the nucleus in connection with DNA-binding transcription factors, YAP (Yes-associated protein), and co-factors like Fast1/FoxH1 and TAZ. (**F**) PPM1A functions as a phosphatase and terminates TGF-β signaling. (**G**) AKT/PKB, ERK/MAP, CDKs, and GSKB3b can inhibit the nuclear accumulation of R-SMADS by phosphorylation with the linker region of R-SMADs.

**Figure 3 pharmaceuticals-17-00326-f003:**
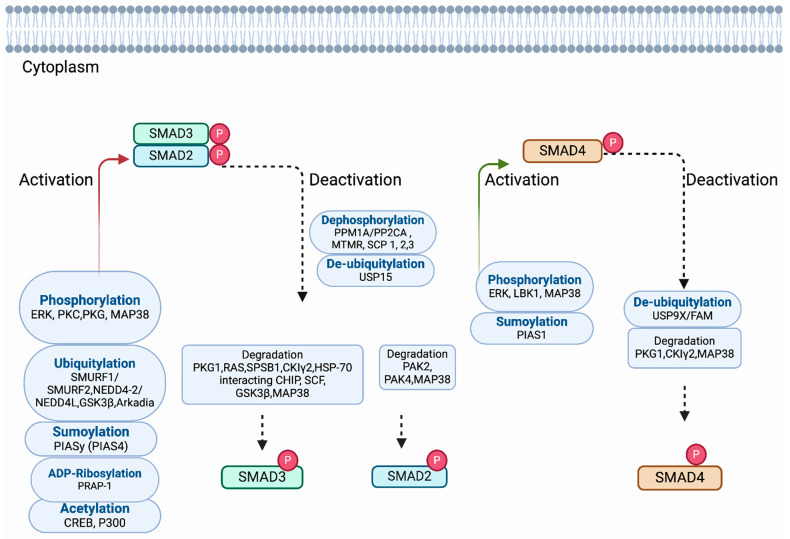
**Regulation of SMADs by post-translational modifications (PTMs).** The complexes of R-SMADs and SMAD4 translocate from the cytoplasm to the nucleus and control the expression of target genes. Here, all the proteins are shown in the cytoplasm. The name of each post-translational modification and the associated proteins involved in each mechanism are designated in the box. A solid red arrow and green arrow indicate that R-SMADs and SMAD4 are modified by activation through post-translational modification. The black dashed arrow indicates the degradation of R-SMADs and SMAD4 and whether R-SMADs and SMAD4 are modified by deactivation and degradation.

**Figure 4 pharmaceuticals-17-00326-f004:**
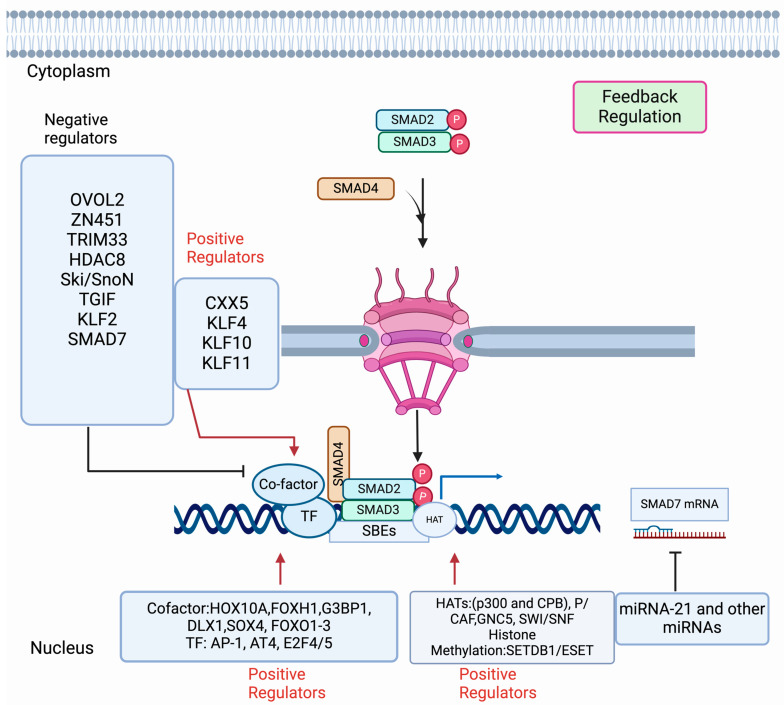
**Regulation of SMAD-mediated transcription**. Activation of R-SMADs (SMAD2, SMAD3) by TGF-β receptor type I promotes the complex formation with R-SMADs-SMAD4. Then the trimeric complexes are imported in the nucleus to turn on or off the target genes. The name of the mechanism and the associated proteins are designated in each box. A solid blue arrow indicates SMAD-mediated transcription. The solid red arrow indicates the regulators that act positively on the SMAD-mediated transcription. The solid black inhibitory line shows the inhibition function mediated by the regulators. Diverse regulators [co-factors, transcription factors (TF), Histone acetyltransferase, methyltransferase, etc.] act on the activated SMAD complexes inside the nucleus and control SMAD-mediated transcription in a context-dependent manner.

**Figure 5 pharmaceuticals-17-00326-f005:**
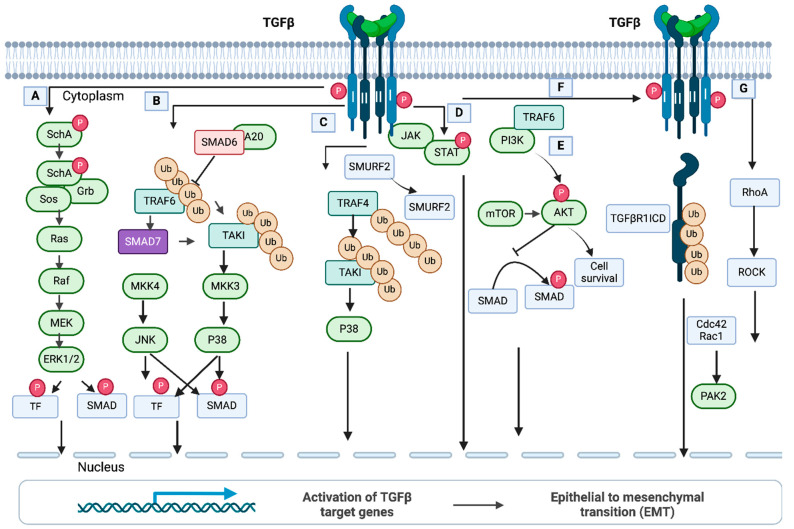
**Non-SMAD, non-canonical TGF-β pathways.** (**A**) ERK/MAP kinase pathway. TGF-β activates the ERK/MAP kinase pathway by direct phosphorylation of ShcA, which subsequently activates downstream signals. In conjugation with SMADs, Erk regulates target gene expression and EMT via the downstream TFs (transcription factors). (**B**) JNK and p38 MAP kinase pathway. The Tumor Necrosis Factor Receptor-Associated Factor 6 (TRAF6) and TGF-β complexes interact after activation of the receptors. TRAF6 is then induced by an autoubiquitylation and subsequently stimulates TAKI through polyubiquitylation. This event can lead to activation of the p38 MAP kinase pathway. (**C**) The MAP kinase pathway via TRAF4 (Tumor Necrosis Factor Receptor-Associated Factor 4). TRAF4 is auto-ubiquitylated upon ligand binding and is recruited to the TGF-β receptor complex. (**D**) JAK/STAT (the Janus Kinase/Signal Transducer and Activator of Transcription) pathway. Phosphorylation and activation of STAT is initiated by JAK, which interacts with TGF-β receptor type I. (**E**) PI3/AKT/mTOR pathway. TRAF6 polyubiquitylates the regulatory subunits of PI3K, p85α. Then, a complex is formed between the TGF-β receptor type I and p85α, resulting in activation of PI3K and AKT. Phosphorylation of AKT prevents SMAD3-mediated signaling. (**F**) TGF-β type I receptor intracellular domain signaling. The interaction of TRF6 and TGF-β receptor I results in polyubiquitination of TGF-β receptor I at Lys63 and subsequent degradation by TNF-alpha converting enzyme (TACE). TACE is not shown here. The newly formed intracellular domain (ICD) of TGFβ receptor I, named TGFβRI ICD, associates with the transcriptional factor to activate genes. (**G**) Rho-like GTPase pathway. Activation of Rho GTPase is promoted by both SMAD and non-SMAD-mediated signaling. After stimulation, Rho-like proteins Cdc2 and Rac1 are activated, which drive actin reorganization through signals from PAK2. This figure has been adapted from Tzavlaki and Moustakas, 2020 [43].

**Figure 6 pharmaceuticals-17-00326-f006:**
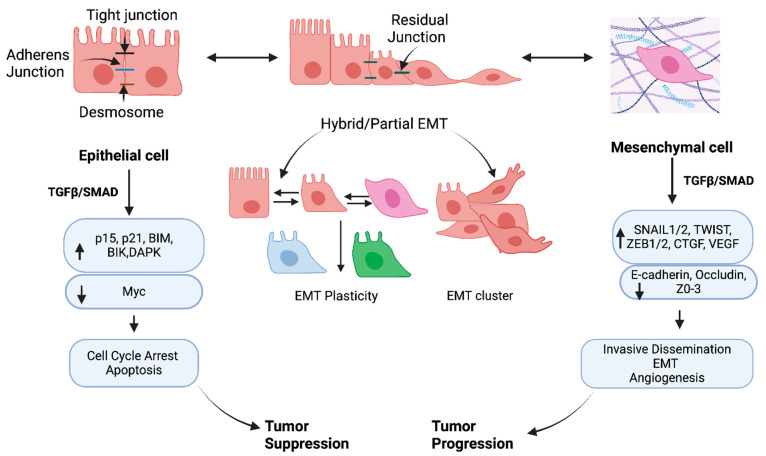
**TGF-β/SMAD mediated tumor suppression and tumor progression in epithelial and mesenchymal cancer cells.** In normal epithelium and pre-malignant cells, TGF-β/SMAD promotes cell cycle arrest and apoptosis via induction of CDK inhibitors (G1 Arrest activating Cyclin-Dependent Kinase) p21 and p15, B-cell lymphoma-2 (BCL-2) family members, BIM (BCL2L11) death-associated kinase (DAPK), and BH3-protein BIK and suppression of c-Myc. At the later stage, TGF-β/SMAD-mediated induction of SNAIL1/2, TWIST, and ZEB1/2 is associated with EMT and tumor progression in mesenchymal cells. The intermediate stage is hybrid/partial EMT, having intermediate polarity and loose intercellular junctions. The partial EMT shows plasticity, which stimulates the cell differentiation and clusters of cells for collective migration. This figure has been adapted from Huang et al., 2022 [289].

**Table 1 pharmaceuticals-17-00326-t001:** Inhibitors of SMAD-interacting enzymes controlling SMAD nucleocytoplasmic trafficking.

Target	Name of Inhibitors	Characteristics/Functions	Preclinical Study In Vitro (Cell Type) and In Vivo Models	Cancer Type	Reference
AKT/PKB (Se/Thr kinase/protein kinase B)	Naltrindole	A classic δ opioid antagonist that reduces cell growth and prompts apoptosis.	NCI-H69, NCI-H345, and NCI-H510	Lung cancer	[69]
	Small-molecule inhibitors of AKT (Aktis): Akti-1/2a, Akti-1, Akti-2, Akti1/2	The inhibitors have pleckstrin homology domain-dependent, isozyme-specific activity. They sensitize tumor cells to apoptotic stimuli.	NCaP, MDA-MB468 A2780, BT474, HT29	Breast cancer	[70]
	GDC-0068	ATP-competitive AKT inhibitor. GDC-0068 inhibits AKT functions, resulting in inhibition of cell cycle progress and cancer cell viability.	MCF7-neo/Her2, BT47M1, PC-3 In vivo broad-spectrum human cancer xenograft models	Multiple solid tumors	[71]
	MK2206	An allosteric AKT inhibitor. MK2206 and gemcitabine inhibit AKT phosphorylation and reduce viability of pancreatic cancer cells.	PANC1, Mia PaCa-2, BxPC-3, AsPC-1, SW1990	Pancreatic cancer	[72]
	MK2206	MK2206 decreases cell proliferation and stemness capacity to form colon spheres and initiate tumor formation.	Human CRC cell line HCT-116 In vivo mouse xenograft	Colorectal cancer	[73]
CDKS (cycline-dependent kinase)	Nitazoxanide (NTZ)	TIZ constrains CDK1 phosphorylation at Thr161 and decreases CDK1/cyclin B1 complex function. TIZ induces the cell cycle arrest at the G2/M phase. In vivo, TIZ reduces the growth of subcutaneous and intracranial orthotopic xenograft models of glioblastoma.	U87, U118, and A172 human glioblastoma cells In vivo mouse xenograft model	Glioblastoma	[74]
	Adapalene (ADA) synthetic retinoid	ADA inhibits cancer cell proliferation, migration, and invasion. ADA reduces tumor growth and bone damage.	RM-1 prostate cancer cell line	Prostate cancer	[75]
	Adapalene (ADA) and combination of PI3K inhibitor (GDC-0941)	The combination of inhibitors showed synergistic effect. Reactive oxygen species accumulation from ADA and GDC caused apoptosis and enhanced sensitivity to GDC in TNBC.	Breast cancer cell lines: MDA-MB-231, MDA-MB-468 MCF-7	Triple-negative breast cancer (TNBC)	[76]
CDKs (2/4)	CDK2i, CDK4i,	Phosphorylation and gene reporter functions of SMAD2 and SMAD3 are reduced by CDK inhibitors with TGF-β stimulation.	MDA-MB-231, MDA-MB-436, and Hs578T cell lines	Breast cancer	[77,78]
CDKs(2.4/6)	Vanoxerine dihydrochloride (CDK2/4/6 triple inhibitor)	Vanoxerine dihydrochloride arrests the cell cycle, induces apoptosis, and produces a synergistic cytotoxic effect in HCC cells. In vivo, tumor growth can be reduced in mouse models.	The human HCC cell lines QGY7703 and Huh7 In vivo mouse model	Hepatocellular carcinome	[79]
GSK3b axin/glycogen synthase kinase 3 beta	Tideglusib, AZD1080, BIO	The inhibitors target GSK3 and associated signaling pathways and modify the phosphorylation of GSK-3 substrates, such as T53 on c-MYC and S33/S37/T41 on β-catenin. Modulation of KRAS-dependent tumor growth is initiated by the inhibitors.	Human lung, colon pancreatic, and prostate cancer cell lines. Calu-6, A549, H460, PC9, and H4006 SW620, DLD-1, and HCT-8 MiaPaCa2 L3.6pl DU145 HEK293 Female athymic nude mice (human tumor xenograft)	Lung cancer Pancreatic cancer Prostate cancer	[80,81]
	The glutamate release inhibitor, Riluzole	Pro-oncogenic function of SMADs is modulated by Riluzole. It increases linker region phosphorylation of SMAD2 and SMAD3 at serine clusters through GSK3.	Melanoma cell lines WM793, WM278, and 1205LU SiRNA GSK3α/β knock-down	Melanoma Cancer	[82]
ERK (extracellular signal-regulated kinase)	INR119	INR119 inhibits MEK1/2 in cancer cells. It can induce ROS by ERK signaling with increased kinase activity. As a result, the proapoptotic genes (TP53, BAX) are highly expressed, resulting in apoptosis.	Human breast cancer cell MCF-7	Breast cancer	[83]

**Table 2 pharmaceuticals-17-00326-t002:** Inhibitors of SMAD-interacting enzymes controlling SMAD post-translational modification.

Target	Name of Inhibitors	Characteristics/Functions	Preclinical Study In Vitro (Cell Type) and In Vivo Models	Cancer Type	Reference
Erk Mapk	MEK/ERK inhibitors	The inhibitor targets MEK/ERK by decreasing serine phosphorylation of SMAD2/3, except phosphorylation of the C-terminal motif.	Human mesangial cells Mouse mammary gland epithelial cells (NMuMGs)		[120]
ERK1/2	FR180204	FR180204 affects cell proliferation, apoptosis, and migration in CRC cells with inhibition of MEK/ERK signaling.	HCT116, Caco-2	Colorectal	[121]
	Thienyl benzenesulfonate scaffold	Thienyl benzenesulfonate scaffold selectively inhibits ERK1/2 substrates that specifically have an F-site or docking site with DEF motif for ERK. It induces apoptosis in melanoma cells containing mutated Braf.	HeLa cervical cancer cell, leukemia cell; Jurkat T cell, melanoma cell; A375 and RPMI7951	Melanoma	[122]
	BI-78D3	BI-78D3 binds DRS (D-recruitment site) of ERK2 by making a covalent bond with a cysteine residue (C15.9) and disrupts TGF-β/SMAD signaling. It induces apoptosis in melanoma cells.	HEK293T, CRL-3216 melanoma cell line: A357, CRL-1619	Melanoma	[123]
	DEL-22379	DEL-22379 inhibits ERK dimerization without affecting its phosphorylation by RAS-ERK pathway, leading to apoptosis and preventing tumor progression.	Human cell lines Mutant BRAF RAS In vivo mouse tumor model		[124]
	Piperine	Piperine reduces the phosphorylation of SMAD2 and ERK1/2 and shows an anti-EMT effect.	A549, MDA-MB-231, and HepG2	Lung cancer	[125]
PKC (protein kinase C)	Chelerythrine	A natural benzophenanthridine alkaloid. Targeting PKC shows a selective antiproliferative effect on TNBC cells.	MDA-MB-231, BT-549, HCC1937, MDA-MB-453, MDA-MB-468 MCF7, ZR-75, and SK-BR3 In vivo xenograft mouse model	Breast cancer	[126]
	BD-15	BD-15 enhances PKC signal and upregulates p21 expression and phosphorylation.	Lung cancer cell lines	Lung cancer	[127]
	Bisindolylmaleimide I, Gö6976, and Rottlerin	The inhibitors induce ROS-induced apoptosis in cancer cells in human colon cancer cells.	CCD18Co and Caco-2 colon adenocarcinoma cells, CCD18Co, normal colon cell	Colon cancer	[128]
PKG1 (Protein kinase G)	SBA (sulindac benzylamine)—a novel sulindac derivative lacking cyclooxygenase (COX)-inhibitory activity)	SBA is known as cyclic guanosine 3′,5′, -monophosphate phosphodiesterase (cGMP PDE). SBA inhibits cGMP hydrolysis in colon tumor cells and activates cGMP-mediated PKG, which suppresses tumor cell growth.	HT-29, SW480, HCT116, and FHC (human fetal colonocytes)	Colon cancer	[129]
	SSA (sulindac sulfide amide—lacks COX-inhibitory activity)	cGMP PDE inhibitor. SSA targets cGMP/PKG and inhibits b-catenin/Tcf transcriptional activity, resulting in apoptosis of breast cancer cells and mammary tumorigenesis in rats.	Hs578t, MCF-7, ZR-75, SKBr3, MDA-MB3-231 In vivo rat model	Breast cancer	[130]
CK1 (Casein kinase 1)	SR3029	SR3029 targets CK1d and upregulates deoxycytidine kinase (dCK). A combination of SR3029 with gemcitabine-induced synergistic antiproliferative and enhanced apoptosis.	Human PDA cell lines: BxPC-3, MIAPaCa-2, PANC1 Bladder cancer cell lines: UM-UC3, TCCSUP, 5637, HT-1376, J82, T24, Orthotopic pancreatic and bladder cancer model in mice	Pancreatic cancer Bladder cancer	[131]
	IC261	IC261 targets CK1(d/e) isoforms and influences colon cancer cell growth and apoptosis by increasing aerobic glycolysis through p53-dependent mechanism.	HCT116, RKO, LOVO, SW480	Colon cancer	[132]
PAK2 (p21 Activated Kinase	FRAX597	FRAX597 is an ATP-competitive, which significantly reduces NF2-deficient Schwann cell growth in vitro and tumor in a xenograft model.	SC4 cells, *Nf2*−/−SC4 Schwann cells In vivo tumor model in mouse	Neurofibromatosis type 2 (NF2)-associated schwannomas	[133]
PAK4	PF3758309 Small-molecule P21-activated kinase inhibitor	ATP-competitive pyrrolopyrazole inhibitor of PAK4-dependent pathway blocked multiple tumor xenografts.	92 tumor cell lines Human xenograft tumor model	Breast cancer Pancreatic cancer Colorectal cancer Non-small-cell lung cancer	[134]
	Compound 31 Compound55	Compound 31 inhibits cell proliferation, migration, and invasion of tumor cells by modulating the PAK4-mediated signaling pathways. Potential in antitumor metastatic efficacy and mitigation of TGF-*β*1-induced epithelial–mesenchymal transition (EMT).	Lung cancer cell A549 and pharmacokinetic assessment in rats Lung cancer cell A549 and melanoma line B16 In vivo zebrafish embryo and mouse model	Lung cancer Lung cancer melanoma	[135,136]
LKB1	Sunitinib	Sunitinib is multitarget angiogenesis. It reduces tumor size and necrosis. Metastatic and nonmetastatic mouse models show an increase in median survival.	KW-634 KW-857 In vivo mouse model	Non-small-cell lung cancer	[137]
	AZD8055/2-DG	A combined treatment of AZD8055/2-DG reduced mammary gland tumorigenesis by inhibiting mTOR pathways and glycolytic metabolism.	Primary, mammary epithelial cells, LKB1^−/−^NIC mice, and wild-type mice	Breast cancer Metastatic lung tumor	[138]
	B-RAF-V600E	Targets LBK-AMK RAF-MEK-ERK signaling, allows activation of AMK, and inhibits melanoma cell proliferation.	K-Mel-28, UACC62, UACC257, SK-Mel-31, and MeWo Cell	Melanoma cancer	[139]
PPM1A/PP2CA Phosphatase	SAMP (small-molecule activators of SAMPs) SMAP-2	SAMPs persistently inhibit MYC expression and MYC transcriptional activity. Cancer cell proliferation is inhibited in vitro. Tumor growth inhibition was observed in vivo. SMAP-2 decreases cellular viability, induces apoptosis, and reduces tumor growth.	Lung cancer cell line H441 Breast cancer cell line: BT-549, MDA-MB-453, MDA-MB-231, SUM149, and HCC1143 In vivo mouse xenograft LNCaP, 22Rv1 in vitro and in vivo mouse model	Non-small-cell lung cancer Breast cancer Castration-resistant prostate cancer	[140,141]
	MicroRNA-487a-3p	MicroRNA-487a-3p binds directly with the 3′UTR of PPMIA phosphatase. It can effectively inhibit the expression of the phosphatase enzyme.	CAL-27, CA8113	Oral squamous cell carcinoma	[142]
	PP2A DT-061	A combination of PP2A DT-061 and MEK inhibitor AZD6224 suppresses p-AKT and MYC. Tumor growth in mouse mode was reduced by the action of the inhibitor.	Cell lines: A549, H460, H358, H441, and H2122 In vivo mouse model	Lung cancer	[143]
SCP(1,2,3)	miRNA-26b	miRNA-26 b has antagonist effect of host gene SCP1.	Rat cardiomyocytes	Cardiac hypertrophy	[144]
	Rabeprazole	Rabeprazole is Scp/TFIIF-interacting CTD phosphatase (Fcp/SCP) family. It binds to the hydrophobic binding pocket of SCPs, a proton pump inhibitor that specifically inhibits SCP1. It regulates irinotecan drug resistance topoisomerase 1 degradation.	Cell lines: HCT116, HT29, DLD1, LoVo. Patient study	Colorectal cancer Gastric cancer	[145,146]
NEDD4-2/NEDD4L	Curcumin	Curcumin promotes glioma cell growth inhibition and induces apoptosis. Glioma cell proliferation, migration, and invasion were reduced with reduced expression of NEED4, Notch1, and pAKT.	SNB19 and A1207	Glioma cancer	[147]
	OSI906	OSI906 targets NEDD4, leading to inhibition of gastric cancer cell proliferation dependent on IGF1/IGF1R signaling pathway.	Human GC cell line: BGC803, MKN45, SGC7901, MKN28 Xenograft nude mouse model Patient data	Gastric cancer	[148]
	Diosgenin	Diosgenin inhibits the expression of NEDD4, resulting in anti-tumor effects (inhibition of cell growth, cell cycle arrest, apoptosis, inhibition of cell migration and invasion) in prostate cancer.	PC-3	Prostate cancer	[149]
HSC70-interacting protein (CHIP)	PES(2-Phenylethylenesulfonamide)	PES selectively interacts with HSP70 and disrupts the interaction of many co-chaperons, substrate proteins, and multiple signaling pathways. This suppresses tumor growth in mouse models.	Transgenic Eμ-*Myc* mouse model of lymphomagenesis	Lymphogenesis	[150]
	Pinaverium bromide	Pinaverium bromide inhibits the intracellular chaperon activity of HSP70 system and elicits cytotoxic activity by activating apoptosis in melanoma cells.	Tumorigenic melanoma cell lines: A2058 and MeWo	Melanoma	[151]
SCF (Skp1, Cullin1 and Fbw1a)/ROC	6-OAP	Binding of SKP1 and 6-OAP regulates the interaction of SKPI-SKP2, resulting in prometaphase arrest.	16HBE, HLF, 293T, and a panel of lung cancer cell lines In vivo murine model	Lung cancer	[152]
	Z0933M	Binds C-terminal of SKP1 and inhibits the association of F-box protein to make stable SCF E3 ligase. It disrupts SCF and induces cell death by p53-dependent mechanism.	A panel of different cancer cells. MDA-231, MCF-7, Hela, BTC6, HEK293, HepG2, HCT and A-431	Breast cancer	[153]
SCFskp2 E3 ligase	C-series compound (C1, C2, C16, C20)	C-series compounds inhibit Cks1 activity to destabilize SKP2-p27 interaction, enhance p27 accumulation, and promote cell type-specific blocks in G1 or G2/M phase.	MCF-7, T47D, 501Mel, SK-MEL-173, SK-MEL-147	Melanoma	[154]
	Dioscin	Dioscin promotes SKP2-CDH1 interaction to induce CDH1-mediated degradation of SKP2 and delays tumor growth.	Colorectal cell line: DLD, HCT116, SW480, HT29, HCT8, SW620 In vivo mouse model	Colorectal cancer	[155]
	Compound 25 (C25)	C25 inhibits interaction of SKP2 with adaptor SKP1 and the ligase activity of SKP2, resulting in cancer progression.	293T, PC3,A549, H460, H1299, Hep3B, U2OS	Liver, lung, prostate, and osteosarcoma	[156]
P300 CREB protein and P300 Histone acetylase)	A-485	A-485 arrested p300/pCBP-mediated histone acetylation marks of cell senescence in NSCLC. It regulates antitumor activity in many solid tumors.	Non-small-cell lung cancer (NSCLC) cell lines: NCI H460, NCI 1650, H1299 PC-3 In vivo mouse xenograft	Non-small-cell lung cancer Prostate cancer Melanoma	[157,158]
	B029-2	B029-2 inhibits glycolysis and induces tumor cell cycle arrest by reducing through modulation of histone acetylation.	Huh7, Hep3B In vivo mouse xenograft	Hepatocellular carcinoma	[159]
	PU141	PU141 is a selective inhibitor to p300/pCBP that reduces tumor growth in vivo through the reduction of histone lysine acetylation.	SK-N-SH neuroblastoma cells In vivo mouse xenograft	Neuroblastoma	[160]
	C646	C646 selectively inhibits p300 and CBP functions. It inhibited cell proliferation and induced apoptosis in vitro.	Human gastric epithelial cells, GES-1 and gastric cancer cell line SGC 7901, MKN45,BGC823, KATOIII	Gastric cancer	[161]
PRAP1	AZD5305	AZD5305 shows anti-proliferative effects in vitro. It potentially and selectively inhibits PRAP1 functions.	In vitro and in vivo mouse xenograft and PDX model. Rat preclinical model cell; MDA-MB-436, DLD-1, DLD BRACA2−/−	Breast cancer	[162]
	[^77^Br]Br-WC-DZ	A radio-brominated Auger emitting inhibitor targeting PARP-1. The inhibitor shows cytotoxicity in prostate cancer cells and promotes DNA damage and cell cycle arrest at G2/M phase.	Prostate cancer cell lines: PC-3, IGR-CaP1 In vivo, prostate cancer xenograft model	Prostate cancer	[163]

**Table 3 pharmaceuticals-17-00326-t003:** Inhibitors of SMAD-interacting enzyme in transcription and post-transcription.

Target	Name of Inhibitors	Characteristics/Functions	Preclinical Study In Vitro (Cell Type) and In Vivo Models	Cancer Type	Reference
HAT Histone Acetyltransferase GNC5/PCAF	PU139 Pan-inhibitor	PU139 inhibits GNC5/PCAF function and triggers caspase-independent cell death. It blocks growth of SK-N-SH neuroblastoma xenografts in mice.	SK-N-SH neuroblastoma cell In vivo mouse xenograft	Neuroblastoma cell	[160]
	GSK983 PROTAC (proteolysis-targeting chimeras)	GSK983 targets GNC5/PCAF and modulates immunity through mediators released by LPS-induced immune cells.	Immune cells		[210]
	Garcinol and curcumin (garcinol derivative LTK4)	Garcinol blocks PCAF by modulating the acetylation of the C-terminal domain of p53 in tumor cells.	MCF7 and osteosarcoma cell lines U2OS and SaOS2	Breast cancer	[211]
SWI/SNF	PROTAC Tool Compound AU5330	The inhibitor targets SWI/SNF complex, simultaneously degrades ATPases SMARC4, SMAR2, and PBRMI, and selectively kills H3.3K27M.	BT245, DIPG-007, DIPG-X*IIIp, H3.3 K27M	Lethal pediatric brain cancer	[212]
	The bromodomain inhibitor-PFI3	SWI/SNF‘s chromatin binding is directly blocked by PF13, resulting in DBS repair defects and alternations in damage checkpoints. As a result, necrosis and senescence increase cell death.	A549, HT29, H460, H1299, and U2OS	Several cancer types	[213]
	BRM and BRG1 inhibitors	The inhibitors target ATPase activity of SWI/SNF complex, downregulate BRM-dependent gene expression, and show antiproliferative activity in a BRG1-mutant lung tumor xenograft model.	In vivo mouse xenograft model	Lung tumor	[214]
SETDB1/ESET	Mithramycin A and mithramycin analog (mithralog) EC8042	The inhibitors suppress the expression of SETDB1 and induce changes at transcriptomic, morphological, and functional levels, leading to antitumor effects.	SK-HI SETDB1 melanoma cell line	Malignant melanoma	[215]
	DZNep (deazaneplanocin)A	DZNep inhibits histone methylations, including H3K27me3 and HCK9me3. The reduced levels of H3K27me3 and H3K9me3 decrease the EZH2 and SETDB1 protein levels in lung cancer cells.	Lung epithelial carcinoma cell A549, H1299, H460	Lung cancer	[216]
	miRNA-621	miRNA-621 could directly target the 3′ UTR of SETDB1. Direct inhibition of SETDB1 further boosts the radiosensitivity of HCC cells.	LO2.HepG2, Smmc-7721, Bel 7404 HCC mouse model	Hepatocellular carcinoma	[217]
RAS Small GTPase	BI-3406	BI-3406 is a SOS1/MEK inhibitor that enables tumor growth in different KRAS-driven tumor models.	NCI-H23, FLAG-SOS1, SOS1, and SOS2-negative cells Patient-derived xenograft study Cell-derived efficacy study in mouse model	KRAS-driven cancer	[218]
	KRAS agonist 533	KRAS-533 binds the GTP/GDP-binding pocket of KRAS. KRA-533 increases KRAS activity and suppresses cell growth in lung cancer patients.	A459 Lung cncer xenograft	Lung cancer	[219]
	Kobe0065 Kobe2602	In vitro and in vivo, the inhibitors show inhibitory effect binding with H-Raf-GTP-c-Raf-1. They induce apoptosis and inhibit cell growth.	NIH 3T3 cells transformed with H-rasG12V SW480 Tumor xenograft	Colorectal cancer	[220]
	ARS-1620	The inhibitor dissects oncogenic KRAS decency.	Subcutaneous xenograft models bearing *KRAS p.G12C.*	NSCLC	[221]
HDAC1 Histone Deacetylase	Romidepsin	HDAC1/2 inhibitor, which suppresses diethylnitrosamine (DEN)-induced hepatocellular carcinoma (HCC). HDAC1 inhibitor. Cells treated with romidepsin showed apoptotic cell death and reduced HDAC activity.	C56BL/6 mice A panel of 8 BTC cell lines	Hepatocellular carcinoma (HCC) Biliary tract cancer (BTC)	[222,223]
	NK-HDAC-1	Cell cycle arrest, apoptosis effects, and inhibition of cell growth were observed.	MDA-MB-231, MCF-7 In vivo mice	Breast cancer	[224]
	CG200745	Inhibition of pancreatic cancer cell growth by overcoming gemcitabine resistance.	BxPC3, Cfpac-1, HPAC Xenograft mouse model	Pancreatic cancer	[225]
		CG200745 causes epigenetic reactivation of critical genes and induces antiproliferation in NSCLC cancer.	Lung cancer cell lines; NSCLC and Beas-2B (Beas-2B)	Lung cancer	[226]
	FR901228	Targets CDKA1A/p21 to induce cell cycle arrest.	In vitro MCF-10A, PC-3, DU145, SW620, IGROV, MCF-7, A549	Many cancer types: breast cancer, prostate cancer, ovarian cancer, colon cancer, and lung cancer	[227,228]
HDAC8	Organoselenium compounds MSC—methyselenocysteine SM—selenomethionine	MSC and SM are HDAC inhibitors (HDAC1 and HDAC8) that generate metabolites from α-keto acid and potentially affect histone and chromatin remodeling.	Human HT29 and HCT116, HCT116(53−/−), HCT116(53+/+)	Colon cancer	[229]
	NCC149 derivatives	A selective inhibitor of HDAC8 that increases a-tubulin acetylation and suppresses T-cell lymphoma cells.	HeLa cells	Cervical cancer	[230]
	HDAC8 selective inhibitors (cpd2, PCI-04051, PCI-48000, PCI-48012) with retinoic acid	In vitro and in vivo, the inhibitors reduce neuroblastoma growth by selective inhibition of HDAC8 with retinoic acid.	Human neuroblastoma cell lines; BE(2)-2,IMR-32, SH-SY5Y, SK-N-AS and SH-EP Mouse xenograft study with HDAC8 knockdown	Neuroblastoma	[231]
m^6^A Methyltransferase	Quercetin (derived from natural products)	Quercetin can inhibit METTL3(methyltransferase complex), decrease m6A level, and inhibit tumor cell proliferation.	MIA PaCa-2	Pancreatic cancer	[232]
	STM2457	A small-molecule inhibitor of METTL3. STM2457 affects the inhibition of catalytic activity and upregulation of METTL3, resulting in upregulation of PD-L1 and reduction of tumor progression.	A panel of lung cancer and lung epithelial cell lines A549, H1975 HBE135, BEAS-2B In vivo mouse model	Non-small-cell lung cancer Oral	[233]
	BI-78D3	BI-78D3 binds DRS (D-recruitment site) of ERK2 by making covalent bond at C159. Apoptosis was induced in different melanoma cell lines, including BRAF inhibitor-naive and resistant melanoma cells.	HEK293T, CRL-3216 Melanoma cell line: A357, CRL-1619	Melanoma	[123]
SET (7/9) Methyltransferase	(*R*)-PFI-2	(*R*)-PFI-2 is a selective inhibitor of SET 7. It can cause modulation of Hippo pathway by increasing nuclear YAP and YAP-mediated gene transcription.	Murine embryonic fibroblasts (MEFs) MCF7 cells	Breast cancer	[234]

**Table 4 pharmaceuticals-17-00326-t004:** Inhibitors of SMAD-interacting enzymes that control noncanonical TGF-β pathways.

Target	Name of Inhibitors	Characteristics/Functions	Preclinical Study In Vitro (Cell Type) and In Vivo Models	Cancer Type	Reference
c-Jun N-terminal kinase (JNK)	SP0016125	Inhibition of JNK and induction of apoptosis by SMAD-mediated caspase activation. Targeting JNK and activating BAX and dihydroartemisinin (DHA)-induced human lung adenocarcinoma cell apoptosis.	The human cholangiocarcinoma cell RBE and PT67 ASTC-a-1, A549	Cholangiocarcinoma Lung adenocarcinoma	[264,265]
	Polyphylin I (PPI)	JNK pathway is targeted by PPI. In glioblastoma cells, G2/M phase arrest and apoptosis are observed. Although the expression of Bax, p-JNK, and cytochromes are upregulated, anti-apoptotic Bcl-2 protein is downregulated.	U251 glioblastoma cell	Glioblastoma	[266]
	AS601245 and clofibrate (PPARa agonist)	Inhibition of JNK pathway. STAT3 signal is reduced.	CoCo-2, HepG2	Colon cancer	[267]
	JNK-in-IX	JNK-in-IX is specific inhibitor against JNK2. It causes DNA damage through G2 arrest mediated by p53 and p21.	ASpC-1, BxPC-3 MIA Paca-2 Human pancreatic organoid	Pancreatic cancer	[268]
	JNK-IN-8	Lapatinib and JNK-IN-8 synergistically inhibit transcriptional activity of AP-1, Nrf2, and NFӄ to promote apoptosis.	MDA-MB-436, HCC1569, MDA-MB-231 In vivo mice	Breast cancer	[269]
p38 MAP kinase	LY2228820	LY2228820 is an ATP-competitive inhibitor of the α- and β-isoforms of p38 MAPK. It inhibited tumor growth in various in vivo cancer models.	A549, U-87MG, HeLa, MDA-MB-468, 786-O, OPM-2, A2780 In vivo mice xenograft model	Melanoma, non-smal—cell lung cancer, ovarian, glioma, myeloma, breast	[270]
	BIRB-796 with VX680	Dual blocking of Aurora kinase and p38 MAPK. Reduced cell proliferation of cervical cancer.	HeLa, Caski, and SiHa Human tumor xenograft in nude mice	Cervical cancer	[271]
	BRIB796	RIB796 blocks G1 phase cycle and inhibits cell proliferation, migration, and migration in GMB cell lines.	U87, U251	Glioblastoma	[272]
AKT	CMG002 and sorafenib	HCC cell proliferation and tumor growth were reduced by inhibition of MAPK and PI3K/AKT/mTOR pathways.	Human HCC cell line, Huh-7, and HepG2	Hepatocellular carcinoma	[273]
	Phycocyanin	Phycocyanin modulates MAPK, Akt/mTOR, and NF-ӄBA pathways to induce apoptosis and autophagic cancer cell death. Complex regulation of the MAPK, Akt/mTOR, and NF-κB signaling pathways.	Pancreatic cancer cell lines: PANC-1, BxPC-3 Other cells: MCF-7, HepG2,HK-2, BGC-823	Pancreatic cancer	[274]
	Quercetin	Inhibition of PI3/AKT and MEK/ERK pathways and induction of apoptosis.	Melanoma B6-F10	Melanoma	[275]
	Silibinin	Triggering the MAP2K1/2-MAPK1/3 pathway but blocking the PI3/AKT/mTOR pathway to induce autophagy and apoptosis.	Colorectal cancer SW 480, HT29, and LoVo cells In vivo mice	Colorectal cancer	[276]
	NVD-LD-225 NVP-BEZ-235	Targets sonic hedgehog and PI3/AKT/mTOR pathways and suppresses tumorigenic potential of glioblastoma initiating cells.	Glioblastoma-initiating cells from patients In vivo mice	Glioblastoma	[277]
TRAF6	TMBPS [bis (4-hydroxy-3,5-dimethylphenyl) sulfone]	TMBPS directly targets TRAF6 to reduce its level. Thereby, it controls multiple pathways like protein kinase B, AKT, and ERK1/2, resulting in cell cycle arrest, apoptosis, and inhibition of tumor growth.	In vitro In vivo mouse xenograft model	Hepatocellular carcinoma	[278]
	Cinchona alkaloids (small-molecule inhibitor, competitive inhibitor of ring domain of TRAF6)	Cinchona alkaloids are potential anti-tumor inhibitors that induce apoptosis both in vitro and in vivo. Ubiquitination and phosphorylation of AKT and TAK1 are inhibited, and Bax/Bcl-2 is upregulated. In vivo, study shows an increase in cytokine production like TNF-α, IFN-γ, and IgG.	HeLa cells	Human cancer	[279]
RhoA Selected small-molecule inhibitor	Rhosin	Inhibition of RhoA activation; blocks GEF binding.	NIH3T3, HME, MCF-7	Breast cancer	[280]
	Y16	Inhibition of RhoA activation by targeting LARG; blocks RhoA binding.	MCF-7, MSF10A	Breast cancer	[281]
	CCG-1423	Inhibition of RhoA activation by targeting MKL1; blocks RhoA-dependent gene transcription.	HK293T, PC-3 NIH3T3	Prostate cancer	[282,283]
	CHS-111	Inhibition of RhoA activation by targeting PLD; blocks RhoA membrane recruitment.	Rat neutrophile		[284]
PKA	PKI (PK inhibitors): PKIA, PKAIB, and PKIG Synthetic peptide analogs of PKI	Alteration of PKA activation, which drives GPCR-Gαs-cAMP signaling toward EPAC-RAP1 and MAPK.	HEK293 Prostate epithelial cell line: RWPE Prostrate adenocarcinoma cell line: LNCaP, VCaP, DU145, and PC3 In vivo mouse	Prostate cancer	[285]
	PKI (6–22) amide	PKI modulates the responses of cancer cells treated by Taxol and Taxane therapeutics. It inhibits the cAMP-PKA pathway in breast cancer cells and reverts the proliferative effect of oxytocin-treated tumors.	Tet-activator expressing LNCaP (LNGK9) and DU145 cells MDA-MB231	Prostate cancer Breast cancer	[286,287]
	PKI (1–25) amide	Cardiac protection through cAMP-dependent EPAC/Rac1/ERK signaling pathway.	Transgenic mouse cardiomyocytes in mice	Cardiomyocytes	[288]

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
