# Peer review of "Targeting SMAD-Dependent Signaling: Considerations in Epithelial and Mesenchymal Solid Tumors"

_pharmaceuticals, 2024, doi:10.3390/ph17030326_

Round 1

Reviewer 1 Report

Comments and Suggestions for Authors

Dear Author/s,

The current review article by you and you team on targeting SMAD4-dependent signaling in solid tumors; however, the article requires major revisions before it can be accepted for publication. Kindly see the points below for the suggestions:

Line 24: Change the word ‘sing’ to ‘since’.

Line 27: Please elaborate ‘transport information’. What do you mean by this?

Line 70: Correct the word ‘TGF-b’ to ‘TGF-β’.

Line 79-89: Figure 1 needs to be edited to add ‘A, B, C, D….’ at respective positions in the pathways. And then please change the caption as well to make it more understandable. For examples: (A) TGF-β ligand binds to a heteromeric complex…. (B)…., (C)….,

Line 90-118: The heading ‘structural features of SMADs’ should be removed from the article along with Figure 2, as detailed structures have been already provided in numerous studies.

Line 149-166: Figure 3 needs to be changed by making the positions of ‘A, B, C, D’ clear and then please change the caption as well to make it more understandable.

Line 178: Provide the name/type of ‘mammalian cell’ here in () as well.

Line 194: Provide the citation right after ‘Yang et al’, not at the end of paragraph.

Line 199: Write the names of a few ‘cancer cells’ in which it is found.

Line 229-230: References in Table 1 for example ‘Mehraj et al.,’ need to be removed as they are already provided in numeric form like ‘[81]’.

Line 247: ‘substrates sites for other kinases…..’ Provide a few examples of kinases in brackets.

Line 264: ‘SMAD4 has been observed in cells….’ Which types of cells are you referring to? Please elaborate.

Line 342: please replace ‘posttranslational modification’ with (PTMs) throughout the articles.

Line 358-360: Correct the references in Table 2 in numeric form like ‘[167]’ only.

Line 358: Correct the word in table 2 ‘HeLa cervical cancer cell Jurkat T-csell leukaemia cells’.

Line 358: Correct the characteristics of ‘miR-487a-3p’ in table 2.

Line 405, 419, 421,471, 472, 476, 479, 489, 552, 562: Correct the words ‘TGFb’ to ‘TGF-β’ in all of these positions.

Line 513, 515, 520, 521, 522: Correct the term ‘miR-621’ to ‘miRNA-621’.

Line 525: Correct the references in Table 3 in numeric form like ‘[214]’ only.

Line 526: One word is missing from this sentence ‘first bioavailable a. small molecule..’ kindly correct this.

Line 564: Correct the figure 5 caption and change the word ‘TGFb’ to ‘TGF-β’.

Line 626: Rewrite this sentence ‘a study by Hamidi demonstrated’ by writing ‘Hamidi et al.,’

Line 721, 723, 724, 734: Correct this ‘TGFβ---induced’ at each mentioned place.

Line 738: Correct the references in Table 4 in numeric form like ‘[272}’ only.

Line 801: Correct the references in ‘extrinsic apoptosis [299—301]’.

Line 812: kindly elaborate the term ‘is called MET’.

Line 873: Correct the following ‘overexpression of TWIST 873 [Yang et al., 2004, Lee et al., 2008]’ according to journal guidelines.

Line 953: Correct the following ‘that TGFB/SMAD’.

Line 967-990: The conclusion needs to be more precise and accurate. Please rewrite the text.

Line 1006: Please correct all the references; there is double numbering at each place.

Best wishes

Comments on the Quality of English Language

English is good.

Reviewer 2 Report

Comments and Suggestions for Authors

This review describes every aspect of SMAD-dependent signaling in great detail and depths.Also preclinical studies that evaluate the efficacy of inhibitors targeting major SMAD-regulating and /or -interacting proteins, particularly enzymes that may play important roles in epithelial or mesenchymal compartments within solid tumors are listed nicely giving the reader a very good overview. Figures and tables are well designed and support the text perfectly. I have no issues at all and believe it is a significant contribution to the field.

Reviewer 3 Report

Comments and Suggestions for Authors

The authors present a very thorough review of the current state of affrais regarding TGF-b and its relationship with EMT and cancer progression. 

I would add a paragraph about the research of piperine as an inhibitor of TGF-induced EMT, as it would bring a spotlight to promising drug develpment research (PMID: 34778375, 28330809, 34730139, 32276474)

Reviewer 4 Report

Comments and Suggestions for Authors

The manuscript discusses the pivotal role of the TGF-β/SMAD signaling pathway in human cancer progression, particularly in solid tumors. The study delves into the intricate mechanisms involving SMAD-dependent canonical and non-canonical signals, providing detailed insights into how the pathway influences different aspects of cancer biology. Moreover, the discussion extends to cancer dissemination and metastasis, detailing the contributions of TGF-β/SMAD signaling in interactions with the tumor microenvironment and its impact on invasion, intravasation, extravasation, and colonization. The conclusion underscores the importance of inhibiting EMT as a potential breakthrough in cancer research and proposes avenues for future research and therapeutic strategies, including small-molecule inhibitors targeting SMAD-interacting enzymes, miRNA, immunotherapy, and combination therapies. Overall, the manuscript provides a comprehensive understanding of TGF-β/SMAD signaling in cancer, its clinical implications, and avenues for future research and therapeutic development. However, the manuscript could include a more explicit discussion of the following topics:

1) limitations and potential biases associated with the presented findings.

2) Discussion of existing or potential therapeutic strategies targeting TGF-β/SMAD signaling and potential marker associated with TGF-β/SMAD in cancer treatment.

Round 2

Reviewer 1 Report

Comments and Suggestions for Authors

Dear Author/s,

This manuscript has a potential for publication. 

Comments on the Quality of English Language

Minor punctuations are required in the manuscript.